# SA-PEF: Step-Ahead Partial Error Feedback for Efficient Federated Learning

**Dawit Kiros Redie**  *dawit.k.redie@ntnu.no*
*Department of Electronic Systems*
*Norwegian University of Science and Technology (NTNU)*
*Trondheim, Norway*

**Reza Arablouei**  *reza.arablouei@csiro.au*
*Commonwealth Scientific and Industrial Research Organisation (CSIRO)*
*Pullenvale, QLD 4069, Australia*

**Stefan Werner**  *stefan.werner@ntnu.no*
*Department of Electronic Systems*
*Norwegian University of Science and Technology (NTNU)*
*Trondheim, Norway*
*Department of Information and Communications Engineering*
*Aalto University*
*Espoo, Finland*

**Reviewed on OpenReview:** *https://openreview.net/forum?id=ejnVWfknCm*

## Abstract

Biased gradient compression with error feedback (EF) reduces communication in federated learning (FL), but under heterogeneous (non-IID) data and local updates, the compression residual can decay slowly. This induces a mismatch between where gradients are evaluated and where the (decompressed) update is effectively applied, often slowing progress in the early rounds. We propose *step-ahead partial error feedback* (SA-PEF), which introduces a tunable step-ahead coefficient $\alpha_r \in [0, 1]$ and previews only a fraction of the residual while carrying the remainder through standard EF. SA-PEF interpolates smoothly between EF ($\alpha_r = 0$) and full step-ahead EF (SAEF; $\alpha_r = 1$). For nonconvex objectives with $\delta$-contractive compressors, we develop a second-moment bound and a residual recursion that yield nonconvex stationarity guarantees under data heterogeneity and partial client participation. With a constant inner stepsize, the bound exhibits the standard $\mathcal{O}\big((\eta \eta_0 TR)^{-1}\big)$ optimization term and an $R$-independent variance/heterogeneity floor induced by biased compression. Our analysis highlights a step-ahead-controlled residual contraction factor $\rho_r$, explaining the observed early-phase acceleration, and suggests choosing $\alpha_r$ near a theory-predicted optimum to balance SAEF's rapid warm-up with EF's long-run stability. Experiments across architectures, datasets, and compressors show that SA-PEF consistently reaches target accuracy in fewer communication rounds than EF.

## 1 Introduction

Modern large-scale machine learning increasingly relies on distributed computation, where both data and compute are spread across many devices. Federated learning (FL) enables model training in this setting without centralizing raw data, enhancing privacy and scalability under heterogeneous client distributions (McMahan et al., 2017; Kairouz et al., 2021). In FL, a potentially vast population of clients collaborates to train a

shared model $w \in \mathbb{R}^d$ by solving

$$w^\star \in \arg\min_{w \in \mathbb{R}^d} f(w) := \frac{1}{K} \sum_{k=1}^{K} f_k(w), \qquad f_k(w) := \mathbb{E}_{z \sim \mathcal{D}_k}\big[\ell(w; z)\big], \tag{1}$$

where $\mathcal{D}_k$ is the (potentially heterogeneous) data distribution at client $k$, $\ell(\cdot)$ is a sample loss (often non-convex), and $K$ is the number of clients. In each synchronous FL round, the server broadcasts the current global model to a subset of clients. These clients perform several steps of stochastic gradient descent (SGD) on their local data and return updates to the server, which aggregates them to form the next global iterate (Huang et al., 2022; Wang & Ji, 2022; Li et al., 2024).

Although FL leverages rich distributed data, it faces two key challenges. The first challenge is communication bottlenecks. Model updates are typically high-dimensional, with millions or even billions of parameters, which makes uplink bandwidth a major constraint (Reisizadeh et al., 2020; Kim et al., 2024; Islamov et al., 2025). This has spurred extensive work on communication-efficient algorithms, including quantization (Seide et al., 2014; Alistarh et al., 2017), sparsification (Stich et al., 2018), and biased compression with error feedback (Beznosikov et al., 2023; Bao et al., 2025). The second challenge is statistical heterogeneity. When client data are non-IID, multiple local updates can cause client models to drift toward minimizing their own objectives, slowing or even destabilizing global convergence (Karimireddy et al., 2020; Li & Li, 2023).

To reduce communication, many methods compress client-server messages using quantization (Alistarh et al., 2017), sparsification (Lin et al., 2018), or sketching (Rothchild et al., 2020). Compressors may be unbiased (e.g., Rand-$k$ (Wangni et al., 2018; Stich et al., 2018)) or biased (e.g., signSGD (Bernstein et al., 2019), Top-$k$ (Lin et al., 2018)), with the latter often delivering better accuracy-communication trade-offs at a given bit budget (Beznosikov et al., 2023). However, naive use of biased compression can introduce a persistent bias, leading to slow or even divergent training (Beznosikov et al., 2023; Li & Li, 2023). Error feedback (EF) addresses this problem by storing past compression errors and injecting them into the next update before compression (Seide et al., 2014). This mechanism cancels the compressor bias and restores convergence guarantees comparable to those of uncompressed SGD, assuming standard smoothness and appropriately chosen stepsizes (Karimireddy et al., 2019; Bao et al., 2025).

EF in federated settings faces two important limitations. First, under highly non-IID data, the residual can align with client-specific gradient directions, inducing cross-client gradient mismatch and slowing early progress (Hsu et al., 2019; Karimireddy et al., 2020). Second, EF retains residual mass until fully transmitted. Once the residual norm is small, communication rounds may be wasted transmitting stale, low-magnitude coordinates rather than fresh gradient signal (Li & Li, 2023).

Step-ahead EF (SAEF) (Xu et al., 2021) mitigates the first issue by *previewing* the residual: before local SGD, each client shifts its model by the current error and optimizes from that offset. This strategy often yields a strong warm-up, as the residual is injected in full every round. However, in FL regimes with non-IID data, multiple local steps, aggressive compression, and partial participation, the full step-ahead variant exhibits late-stage plateaus and larger gradient-mismatch spikes compared to EF. Moreover, prior analysis (Xu et al., 2021) has largely focused on classical distributed optimization, leaving open whether one can *systematically* combine SAEF's fast initial progress with the long-term stability of EF in federated settings with local steps and data heterogeneity.

**Our approach: SA-PEF.** We propose *step-ahead partial error feedback* (SA-PEF), which introduces a tunable coefficient $\alpha_r \in [0, 1]$ and applies a *fractional* residual preview, $w_{r+\frac{1}{2}} = w_r - \alpha_r e_r$, while carrying the remaining residual through standard error feedback. This partial preview yields three benefits:

- *Early acceleration with controlled noise.* A moderate (or decaying) $\alpha_r$ reduces the early-round mismatch induced by biased compression, while the preview perturbation naturally diminishes as $\|e_r\|$ contracts.
- *Sharper residual recursion.* We obtain a per-round residual contraction factor

$$\rho_r = (1 - \tfrac{1}{\delta})\big[2(1 - \alpha_r)^2 + 24\,\alpha_r^2(\eta_r L T)^2\big],$$

which is strictly smaller than the EF baseline $2(1 - \tfrac{1}{\delta})$ for a broad range of $\alpha_r$ when the local-work scale $s_r = \eta_r L T$ is small.

- *Graceful interpolation.* SA-PEF recovers EF at $\alpha_r = 0$ (maximal stability) and SAEF at $\alpha_r = 1$ (maximal preview), enabling practitioners to tune the method to the heterogeneity and compression regime.

**Contributions.**

- *Algorithm.* We introduce SA-PEF, a lightweight drop-in variant of Local-SGD under biased compression, with a tunable step-ahead coefficient $\alpha_r$ and compatibility with any $\delta$-contractive compressor.
- *Theory.* We provide a convergence analysis of SA-PEF for nonconvex federated optimization with local updates and partial participation. Our results include new bounds for local drift, key second-moment terms, and a residual recursion; these recover EF as the special case $\alpha_r = 0$ and quantify how step-ahead changes residual memory and drift. As a consequence, we obtain nonconvex stationarity guarantees of order $\mathcal{O}\big((\eta\,\eta_0 TR)^{-1}\big)$ under a constant inner stepsize, where compression enters through $(1 - 1/\delta)$ and the maximal residual contraction $\rho_{\max} := \sup_r \rho_r$.

## 2  Related work

**Error feedback and compressed optimization.** Error feedback (EF) was first introduced as a practical heuristic for 1-bit SGD (Seide et al., 2014) and later formalized as a *memory* mechanism in sparsified or biased SGD (Stich et al., 2018; Karimireddy et al., 2019). By accumulating the compression residual and injecting it into subsequent updates, EF restores descent directions and admits convergence guarantees for broad classes of *contractive (possibly biased)* compressors. These results include linear rates in the strongly convex setting and standard stationary-point guarantees in the nonconvex case (Gorbunov et al., 2020; Beznosikov et al., 2023). More recently, EF21 (Richtárik et al., 2021) and its extensions (Fatkhullin et al., 2025) provide a modern error-feedback framework for compressing full gradients (or gradient differences) at a shared iterate in synchronized data-parallel training with $T{=}1$, achieving clean contraction guarantees and removing the error floor. However, these works assume no local steps and no client drift. A complementary line of work replaces residual memory with control variates (global gradient estimators), as in DIANA and MARINA (Mishchenko et al., 2025; Gorbunov et al., 2021), which reduce or remove compressor bias without maintaining a full residual vector. *EControl* (Gao et al., 2024) regulates the strength of the feedback signal and fuses residual and estimator updates into a single compressed message, providing fast convergence under arbitrary contractive compressors and heterogeneous data.

**Local updates in FL.** Local or periodic averaging (a.k.a. Local-SGD) reduces communication rounds by performing $T > 1$ local steps between synchronizations (Stich, 2019). While effective in homogeneous settings, non-IID data induces *client drift*, where model trajectories diverge across clients, degrading both convergence speed and final accuracy. Several approaches mitigate drift while retaining the communication savings of local updates. *Proximal regularization* (FedProx) stabilizes local objectives by penalizing deviation from the current global model (Li et al., 2020). *Control variates* (SCAFFOLD) estimate and correct the client-specific gradient bias caused by heterogeneity, yielding tighter convergence with multiple local steps (Karimireddy et al., 2020). *Dynamic regularization* (FedDyn) further aligns local stationary points with the global objective via a round-wise correction term, improving robustness on highly non-IID data (Acar et al., 2021).

**Compression with local updates.** Combining local updates with message compression compounds communication savings but also amplifies distortions from both local drift and compression error. FedPAQ (Reisizadeh et al., 2020) performs $T > 1$ local steps and transmits quantized model deltas at synchronization points, exposing explicit trade-offs among the local period, stepsizes, and quantization accuracy. QSparse-Local-SGD (Basu et al., 2019) extends this to contractive compressors, transmitting Top-$k$ updates after $T$ local steps. While achieving significant traffic reduction, it also reveals that aggressive sparsity can destabilize convergence. CSER (Xie et al., 2020) mitigates this with *error reset*, which immediately injects the residual back into the local model to restore stability under high compression. In the federated local-SGD setting with partial participation and biased compression, Fed-EF (Li & Li, 2023) provides a first nonconvex analysis of classical EF and serves as the EF-style backbone that SA-PEF builds upon. On the control-variate side, Scaffnew/ProxSkip (Mishchenko et al., 2022) is a more recent local-training method in the SCAFFOLD family, using probabilistic local updates to achieve theoretical acceleration. However, it still

relies on full-precision exchanges and no inherent compression. CompressedScaffnew (Condat et al., 2022) extends this mechanism with quantization, TAMUNA (Condat et al., 2023) further handles partial client participation, and LoCoDL (Condat et al., 2025) generalizes the analysis to arbitrary unbiased compressors. These methods primarily establish accelerated convergence in (strongly) convex settings. In parallel, SCAL-LION/SCAFCOM (Huang et al., 2024) combines SCAFFOLD-style control variates with compression and, in SCAFCOM, local momentum to handle heterogeneity and partial participation, at the cost of additional per-client state.

**Step-ahead error feedback.** SAEF (Xu et al., 2021) addresses *gradient mismatch*, i.e., the discrepancy between the model used for gradient computation and the model actually updated when delayed residuals are applied. SAEF performs a *preview* shift of the model using the residual before local SGD and augments this with occasional *error averaging* across workers. Although this reduces mismatch and accelerates early progress, the analysis is developed for classical distributed settings with single-step synchronization and *bounded-gradient* assumptions, and does not cover federated regimes with multiple local steps, non-IID data, or biased compressors. Moreover, error averaging requires extra coordination and communication, which is often impractical in cross-device FL.

Despite progress, it remains unclear how to (i) control gradient mismatch in federated settings with local steps and biased compression *without* extra communication, or (ii) combine step-ahead correction with EF to balance early acceleration and long-term stability. Our work closes this gap by introducing SA-PEF, which performs a controlled step-ahead shift with partial residual retention on top of Fed-EF and provides a contraction-based analysis yielding nonconvex guarantees under heterogeneous data.

## 3 Proposed Algorithm

**Motivation: residual-induced mismatch under biased EF.** With *biased* compressors (e.g., Top-$k$), EF maintains a residual $e_r^{(k)}$ of *unsent* coordinates at each client. As a result, although stochastic gradients are evaluated at the received global model $w_r$, the realized update can be viewed as being applied closer to an effective residual-corrected point

$$\tilde{w}_r := w_r - d_r,$$

where $d_r$ denotes the EF carry; heuristically, for biased sparsifiers, it is often close to the mean residual $\bar{e}_r := \frac{1}{K} \sum_{k=1}^{K} e_r^{(k)}$ across clients. This creates a mismatch akin to one-step staleness: gradients are evaluated at $w_r$, while the update direction is influenced by $\tilde{w}_r$. To quantify this effect, consider the *local displacement term*

$$\frac{1}{K} \sum_{k=1}^{K} \mathbb{E} \big\| \nabla f_k(\tilde{w}_r; \zeta) - \nabla f_k(w_r; \zeta') \big\|^2,$$

where $\zeta, \zeta'$ are independent stochastic samples. Under $L$-smoothness and bounded variance, this term is bounded by $\mathcal{O}\big(L^2 \|d_r\|^2 + \sigma^2\big)$. Thus a large residual carry $d_r$ directly increases the mismatch between the point where gradients are computed and the point where the compressed update is effectively realized. This matters because the same displacement effect contributes to the second moment of the accumulated local update $g_r^{(k)}$ (Lemma 3), and therefore propagates into the residual recursion (Lemma 4) and the final stationarity bound. SA-PEF is designed to reduce this mismatch by previewing a fraction $\alpha_r e_r^{(k)}$ of the residual before local optimization, while retaining $(1 - \alpha_r) e_r^{(k)}$ in the EF recursion. The choice of $\alpha_r$ thus controls a trade-off between displacement reduction and EF-style residual stabilization, which our analysis in Section 4 makes precise.

**Algorithm overview.** SA-PEF (Alg. 1) modifies Local-SGD with a *step-ahead preview* and *partial error-feedback*. At the start of round $r$, each client shifts its local model by a fraction $\alpha_r$ of its residual, runs $T$ local SGD steps from the shifted point, and forms the accumulated local update. It then blends the new update with the remaining residual $(1 - \alpha_r) e_r^{(k)}$, compresses the resulting vector, sends the compressed message to the server, and updates its residual as the compression error. The server averages received compressed messages and applies the update to produce the next global model, which is broadcast to all clients.

---

**Algorithm 1:** Step-Ahead Partial Error-Feedback (SA-PEF) for Efficient FL

---

**Input:** rounds $R$, clients $K$, local steps $T$, stepsizes $\{\eta_r\}$, server stepsize $\eta$, step-ahead schedule
$\{\alpha_r\} \subset [0, 1]$, compressor $\mathcal{C}$, initial model $w_0$

**1 foreach** *client* $k = 1, \ldots, K$ in parallel **do**

**2**     $w_0^{(k)} \leftarrow w_0; \quad e_0^{(k)} \leftarrow 0$

**3 for** $r \leftarrow 0$ **to** $R - 1$ **do**

**4**     **foreach** *client* $k = 1, \ldots, K$ in parallel **do**

       /* step-ahead start                                                    */

**5**        $w_{r+\frac{1}{2},0}^{(k)} \leftarrow w_r^{(k)} - \alpha_r e_r^{(k)}$

       /* local SGD ($T$ steps)                                       */

**6**        **for** $t \leftarrow 0$ **to** $T - 1$ **do**

**7**           $w_{r+\frac{1}{2},t+1}^{(k)} \leftarrow w_{r+\frac{1}{2},t}^{(k)} - \eta_r \nabla f_k\big(w_{r+\frac{1}{2},t}^{(k)}; \zeta_{r,t}^{(k)}\big)$

       /* accumulated local update                                 */

**8**        $g_r^{(k)} \leftarrow w_{r+\frac{1}{2},0}^{(k)} - w_{r+\frac{1}{2},T}^{(k)}$

       /* partial EF composition + compression                       */

**9**        $u_{r+1}^{(k)} \leftarrow (1 - \alpha_r)e_r^{(k)} + g_r^{(k)}$

**10**       $c_{r+1}^{(k)} \leftarrow \mathcal{C}\big(u_{r+1}^{(k)}\big)$

**11**       $e_{r+1}^{(k)} \leftarrow u_{r+1}^{(k)} - c_{r+1}^{(k)}$

**12**       **send** $c_{r+1}^{(k)}$ to server

     /* server-side aggregation                                         */

**13**     $\bar{C}_{r+1} \leftarrow \frac{1}{K} \sum_{k=1}^{K} c_{r+1}^{(k)}$

**14**     $w_{r+1} \leftarrow w_r - \eta \bar{C}_{r+1}$

**15**     **broadcast** $w_{r+1}$ to all clients

**16**     **foreach** *client* $k$ **do**

**17**       $w_{r+1}^{(k)} \leftarrow w_{r+1}$

---

**Relation to EF and SAEF.** The step-ahead coefficient $\alpha_r$ interpolates between prior methods: $\alpha_r = 0$ recovers Fed-EF / classical EF in the federated Local-SGD setting (Li & Li, 2023); $\alpha_r = 1$ yields a full step-ahead variant analogous to SAEF (Xu et al., 2021); and $0 < \alpha_r < 1$ previews part of the residual while retaining EF memory through $(1 - \alpha_r)e_r^{(k)}$. Each method admits an averaged residual recursion of the form

$$\bar{E}_{r+1} \leq \rho_r \bar{E}_r + B_r, \qquad \bar{E}_r := \frac{1}{K} \sum_{k=1}^{K} \|e_r^{(k)}\|^2,$$

where $B_r$ collects the terms depending on the gradient norm, data heterogeneity, and stochastic-gradient variance (see Lemma 4 for the explicit form), and the per-round contraction factor $\rho_r$ varies across methods. Table 1 makes this distinction explicit: the three methods differ only in how the residual is split between a preview shift and the EF carry, with the resulting contraction factors listed side by side.

**Residual contraction.** As shown in the third row of Table 1, the SA-PEF contraction factor $\rho_r$ is strictly smaller than EF's value $2(1 - 1/\delta)$ over a broad range of $\alpha_r$ when $s_r = \eta_r LT$ is small (cf. Prop. 1). This improved contraction helps explain the faster early-phase convergence observed under aggressive compression, while the retained factor $(1 - \alpha_r)e_r^{(k)}$ preserves an EF-style residual carry.

Table 1: Unified view of EF, the full step-ahead endpoint, and SA-PEF in federated local-SGD. All three methods share the same server update $w_{r+1} = w_r - \eta\bar{\mathcal{C}}_{r+1}$ and differ only in how the residual $e_r^{(k)}$ is used between rounds. The contraction factor refers to the per-round factor in the residual recursion of Lemma 4, with $s_r := \eta_r LT$.

|  | EF $(\alpha_r = 0)$ | Full step-ahead $(\alpha_r = 1)$ | SA-PEF $(0 < \alpha_r < 1)$ |
|---|---|---|---|
| Local start | $w_{r+\frac{1}{2},0}^{(k)} = w_r$ | $w_{r+\frac{1}{2},0}^{(k)} = w_r - e_r^{(k)}$ | $w_{r+\frac{1}{2},0}^{(k)} = w_r - \alpha_r e_r^{(k)}$ |
| Residual composition | $u_{r+1}^{(k)} = e_r^{(k)} + g_r^{(k)}$ | $u_{r+1}^{(k)} = g_r^{(k)}$ | $u_{r+1}^{(k)} = (1-\alpha_r)e_r^{(k)} + g_r^{(k)}$ |
| Contraction factor $\rho_r$ | $2\left(1 - \frac{1}{\delta}\right)$ | $\left(1 - \frac{1}{\delta}\right)24\,s_r^2$ | $\left(1 - \frac{1}{\delta}\right)\left[2(1-\alpha_r)^2 + 24\alpha_r^2 s_r^2\right]$ |

## 4 Convergence Analysis

### 4.1 Setup and assumptions

**Definition 1** ($\delta$-contractive compression operator). *A (possibly randomized) mapping $\mathcal{C} : \mathbb{R}^d \to \mathbb{R}^d$ is called a compression operator if there exists $\delta \geq 1$ such that, for all $w \in \mathbb{R}^d$,*

$$\mathbb{E}\left[\|\mathcal{C}(w) - w\|_2^2\right] \leq \left(1 - \frac{1}{\delta}\right)\|w\|_2^2, \tag{2}$$

*where the expectation is taken over the internal randomness of $\mathcal{C}$.*

This class includes widely used compressors such as Top-$k$, scaled Rand-$k$, and various quantization schemes. The parameter $\delta$ controls the contraction strength: larger $\delta$ corresponds to weaker contraction and thus more aggressive compression.

Our objective is to establish nonconvex convergence guarantees by upper-bounding the average squared gradient norm $\frac{1}{R}\sum_{r=0}^{R-1}\mathbb{E}\|\nabla f(w_r)\|^2$, where the expectation is over mini-batch sampling, client randomness, and compression.

We adopt the standard assumptions used in (Alistarh et al., 2018; Li & Li, 2023; Beznosikov et al., 2023) and follow the notation of Alg. 1. Fix a communication round $r$ and let $\{\mathcal{F}_{r,t}\}_{t \geq 0}$ denote the natural filtration generated by all randomness up to the beginning of local step $t$ in round $r$. At local step $t \in \{0, \ldots, T-1\}$ on client $k$, the stochastic gradient is

$$g_{r,t}^{(k)} = \nabla f_k\left(w_{r+\frac{1}{2},t}^{(k)}\right) + \xi_{r,t}^{(k)},$$

where $\xi_{r,t}^{(k)}$ is the stochastic noise.

**Assumption 1** (Smoothness). *Each local objective $f_k$ is differentiable with $L$-Lipschitz gradient: $\|\nabla f_k(y) - \nabla f_k(x)\| \leq L\|y - x\|, \ \forall x, y \in \mathbb{R}^d$.*

**Assumption 2** (Stochastic gradients). *For every client $k$ and step $t$, we have: (i) $\mathbb{E}[\xi_{r,t}^{(k)} \mid \mathcal{F}_{r,t}] = 0$, (ii) $\mathbb{E}[\|\xi_{r,t}^{(k)}\|^2 \mid \mathcal{F}_{r,t}] \leq \sigma^2$, and (iii) conditional on $\mathcal{F}_{r,0}$, the family $\{\xi_{r,t}^{(k)} : k = 1, \ldots, K, \ t = 0, \ldots, T-1\}$ is independent.*

**Assumption 3** (Gradient dissimilarity). *There exist constants $\beta^2 \geq 1$ and $\nu^2 \geq 0$ such that*

$$\frac{1}{K}\sum_{k=1}^{K}\|\nabla f_k(x)\|^2 \leq \beta^2 \|\nabla f(x)\|^2 + \nu^2, \qquad \forall x \in \mathbb{R}^d.$$

**Residual recursion and key constants.** A central step in the analysis is to control the accumulated compression residuals. Let $\bar{E}_r := \frac{1}{K}\sum_{k=1}^{K}\|e_r^{(k)}\|^2$ denote the averaged residual energy. Lemma 4 shows that, under a $\delta$-contractive compressor, $\bar{E}_r$ satisfies the recursion

$$\bar{E}_{r+1} \leq \rho_r \bar{E}_r + \left(1 - \frac{1}{\delta}\right)\left[B_r^{(\nabla)} + B_r^{(\nu,\sigma)}\right],$$

with

$$\rho_r = \left(1 - \tfrac{1}{\delta}\right)\left(2(1 - \alpha_r)^2 + 24\,\alpha_r^2 s_r^2\right), \qquad s_r := \eta_r LT, \qquad \rho_{\max} := \sup_r \rho_r.$$

This recursion is the main place where $\alpha_r$, $s_r$, and $\delta$ enter. The theorem below combines it with a descent argument to bound the average gradient norm.

### 4.2 Main convergence guarantee

We now state a stationary-point bound for SA-PEF (full proof in Appendix A.1).

**Theorem 1** (Stationary-point bound with constant inner-loop step). *Assume 1–3 and let the compressor satisfy Definition 1 with parameter $\delta \geq 1$. Run SA-PEF for $R \geq 1$ rounds with constant client stepsize $\eta_r \equiv \eta_0$, and set $s_0 := \eta_0 LT \leq \tfrac{1}{8}$. Suppose further that $18\,\beta^2 s_0^2 \leq \tfrac{1}{8}$ and $\eta \leq \frac{1}{256\,\beta^2 L\eta_0 T}$. Define $\rho_{\max} := \sup_r \rho_r < 1$ and, for each round $r$, define the residual-energy coefficient*

$$\mathcal{E}_r := \eta\,\alpha_r^2\left(\tfrac{1}{\eta_0 T} + \tfrac{3}{2}\eta_0 L^2 T\right) + L\eta^2\left(2\alpha_r^2 + 24\,\alpha_r^2\eta_0^2 L^2 T^2\right) + \tfrac{L\eta^2}{2}, \tag{3}$$

*and let $\mathcal{E}_{\max} := \sup_r \mathcal{E}_r$. Define the effective error constant*

$$\Theta := \frac{16}{\eta} \cdot \frac{\mathcal{E}_{\max}}{1 - \rho_{\max}}\left(1 - \tfrac{1}{\delta}\right)\beta^2\left(8\,\eta_0 T + 288\,L^2\eta_0^3 T^3\right).$$

*If $\Theta \leq \tfrac{1}{2}$, then with $f^\star := \inf_x f(x)$ and initial residuals $e_0^{(k)} \equiv 0$,*

$$\frac{1}{R}\sum_{r=0}^{R-1}\mathbb{E}\|\nabla f(w_r)\|^2 \;\leq\; \frac{32\,(f(w_0) - f^\star)}{\eta\,\eta_0 T\,R} \;+\; \left(1 - \tfrac{1}{\delta}\right)\left[C_\sigma\,\eta_0^2 L^2 T\,\sigma^2 + C_\nu\,\eta_0^2 L^2 T^2\,\nu^2\right] \;+\; \frac{128\,L\,\eta_0}{K}\,\sigma^2,$$

*where*

$$C_\sigma = \frac{32}{\eta}\left[6\,\eta\,\eta_0^2 L^2 T + 96\,L^3\eta\,\eta_0^3 T^2\right] + \frac{32}{\eta}\cdot\frac{\mathcal{E}_{\max}}{1 - \rho_{\max}}\left[4\,\eta_0 + 96\,L^2\eta_0^3 T^2\right],$$

$$C_\nu = \frac{32}{\eta}\left[84\,\eta\,\eta_0^2 L^2 T^2 + 1344\,L^3\eta\,\eta_0^3 T^3\right] + \frac{32}{\eta}\cdot\frac{\mathcal{E}_{\max}}{1 - \rho_{\max}}\left[8\,\eta_0 T + 1344\,L^2\eta_0^3 T^3\right].$$

**Corollary 1** ($\varepsilon$-stationarity and communication complexity). *Under the assumptions and step-size conditions of Theorem 1, let $\widehat{w}$ be sampled uniformly at random from $\{w_0, \ldots, w_{R-1}\}$ and define*

$$\Gamma(\eta_0) := \left(1 - \tfrac{1}{\delta}\right)\left[C_\sigma\,\eta_0^2 L^2 T\,\sigma^2 + C_\nu\,\eta_0^2 L^2 T^2\,\nu^2\right] + \frac{128\,L\,\eta_0}{K}\,\sigma^2.$$

*If $\Gamma(\eta_0) \leq \varepsilon/2$ and*

$$R \;\geq\; \frac{64\,(f(w_0) - f^\star)}{\eta\,\eta_0 T\,\varepsilon},$$

*then $\mathbb{E}\|\nabla f(\widehat{w})\|^2 \leq \varepsilon$. Consequently, the number of communication rounds sufficient to reach $\varepsilon$-stationarity scales as*

$$R = \mathcal{O}\left(\frac{f(w_0) - f^\star}{\eta\,\eta_0 T} \cdot \frac{1}{\varepsilon}\right), \qquad \text{and total stochastic gradients } \mathcal{O}(KTR).$$

**Remark 1** (Typical scaling under the maximal server step). *A common choice is to take the maximal server step allowed by Theorem 1, i.e., $\eta = \frac{1}{256\,\beta^2 L\eta_0 T}$. Then $\eta\,\eta_0 T = \frac{1}{256\,\beta^2 L}$ and the optimization term in Theorem 1 becomes $\frac{8192\,\beta^2 L(f(w_0) - f^\star)}{R}$, independent of $\eta_0$. Choosing $\eta_0$ so that $\Gamma(\eta_0) = \mathcal{O}(\varepsilon)$ (e.g. $\eta_0 = \Theta(\varepsilon)$ up to problem-dependent constants) yields the communication complexity $R = \mathcal{O}(\beta^2 L(f(w_0) - f^\star)/\varepsilon)$.*

### 4.3 Discussion and extensions

**Interpretation of Theorem 1.** Theorem 1 exhibits the standard nonconvex behavior under *biased* compression: with a constant inner stepsize $\eta_0$, the optimization term decays as $\mathcal{O}((\eta\,\eta_0 TR)^{-1})$, while an $R$-independent error floor remains due to residual accumulation. Only the mini-batch variance term averages down with the number of clients (yielding the $1/K$ factor), whereas the residual-induced contribution need not. The floor scales with the local-work budget through $\eta_0^2 L^2$ and grows with the number of local steps ($T$ for stochastic noise and $T^2$ for heterogeneity), in line with earlier analyses of compressed federated optimization (Li & Li, 2023; Karimireddy et al., 2020). Compression affects the bound through the bias factor $(1 - 1/\delta)$ and the residual contraction $\rho_{\max} < 1$.

**Where the $\alpha$-trade-off enters.** For constant $\alpha_r \equiv \alpha$ and $\eta_r \equiv \eta_0$, the dependence on $\alpha$ enters only through (i) the residual contraction $\rho(\alpha)$ in Lemma 4, and (ii) the residual-energy coefficient $\mathcal{E}(\alpha)$ in the descent recursion. In particular, $\mathcal{E}(\alpha) = \frac{L\eta^2}{2} + \alpha^2 \tilde{\mathcal{E}}(\eta, \eta_0, L, T)$ is monotone in $\alpha^2$, whereas $\rho(\alpha)$ can decrease sharply for $\alpha > 0$ when $s_0 = \eta_0 LT \ll 1$. Consequently, the effective residual contribution in Theorem 1 is governed by the ratio $\mathcal{E}_{\max}/(1 - \rho_{\max})$: step-ahead can improve constants by reducing $\rho_{\max}$, but overly large $\alpha$ increases $\mathcal{E}_{\max}$ and may weaken contraction when $s_0$ is not small.

**Why step-ahead helps (and how much).** Step-ahead changes the residual contraction factor to

$$\rho_r = (1 - \tfrac{1}{\delta})\Big(2(1 - \alpha_r)^2 + 24\,\alpha_r^2(\eta_r LT)^2\Big) = (1 - \tfrac{1}{\delta})\Big(2 - 4\alpha_r + (2 + 24s_r^2)\alpha_r^2\Big), \qquad s_r := \eta_r LT \le \tfrac{1}{8}.$$

This quadratic in $\alpha_r$ is minimized at $\alpha_r^\star = \frac{1}{1+12s_r^2} \in (0.84, 1]$, so for small $s_r$ the smallest contraction factor is attained by a *moderate-to-large* step-ahead (indeed, $\alpha_r^\star \to 1$ as $s_r \to 0$). Relative to EF ($\alpha_r = 0$), the contraction strictly improves whenever

$$\rho_r - \rho_{\text{EF}} = (1 - \tfrac{1}{\delta})\big[-4\alpha_r + (2 + 24s_r^2)\alpha_r^2\big] < 0, \qquad \alpha_r \in \Big(0, \frac{2}{1 + 12s_r^2}\Big),$$

and the minimum value is

$$\rho_{\min} = (1 - \tfrac{1}{\delta})\Big(2 - \frac{2}{1 + 12s_r^2}\Big) = \rho_{\text{EF}}\Big(1 - \frac{1}{1 + 12s_r^2}\Big).$$

The trade-off is that increasing $\alpha_r$ also increases the residual-energy coefficient $\mathcal{E}_r$ (scaling as $\alpha_r^2$), and may increase $\rho_r$ through the $\alpha_r^2 s_r^2$ term when $s_r$ is not small. Thus, in the aggressively compressed regime (large $\delta$, hence larger $(1 - 1/\delta)$), maintaining $\rho_{\max} < 1$ requires sufficiently small local work $s_r = \eta_r LT$, or less aggressive compression (smaller $\delta$, i.e., stronger contraction). Overall, step-ahead improves residual contraction without changing the leading optimization rate $\mathcal{O}((\eta\eta_0 TR)^{-1})$; its impact is through residual-dependent constants, most notably via the ratio $\mathcal{E}_{\max}/(1-\rho_{\max})$. Intuitively, a smaller $\rho_r$ means that residual energy decays faster from round to round, so less stale compression error is carried into future updates. This directly reduces the mismatch between the point where gradients are evaluated and the point where the compressed update is effectively realized. This mechanism helps explain the communication-efficiency gains observed in Section 5, and also clarifies why values of $\alpha_r$ near $\alpha_r^\star$, rather than the EF or full step-ahead endpoints, deliver the best empirical performance.

**Partial participation.** Let $m \in \{1, \ldots, K\}$ clients participate in each round, sampled uniformly at random, and define $p := m/K \in (0, 1]$. Suppose the server aggregates the sample mean over participating clients, $\bar{C}_{r+1} = \frac{1}{m} \sum_{k \in \mathcal{S}_r} C(u_{r+1}^{(k)})$ with $|\mathcal{S}_r| = m$. Then $\mathbb{E}[\bar{C}_{r+1} \mid w_r]$ is an unbiased estimator of the full-client mean. Hence, the descent part of the proof remains unchanged, while the pure noise-averaging term is modified by replacing the factor $1/K$ with $1/m$.

In particular, under the assumptions and step-size conditions of Theorem 1, a representative constant-step bound becomes

$$\frac{1}{R} \sum_{r=0}^{R-1} \mathbb{E}\|\nabla f(w_r)\|^2 \le \frac{32\,(f(w_0) - f^\star)}{\eta\,\eta_0 T\,R} + \Big(1 - \tfrac{1}{\delta}\Big)\Big[C_\sigma\,\eta_0^2 L^2 T\,\sigma^2 + C_\nu\,\eta_0^2 L^2 T^2\,\nu^2\Big] + \frac{128\,L\,\eta_0}{m}\,\sigma^2, \quad (4)$$

where $C_\sigma$ and $C_\nu$ are as in Theorem 1 (and thus include $\mathcal{E}_{\max}/(1 - \rho_{\max})$). More refined bounds can additionally capture client-sampling variance beyond the $1/m$ averaging, but this does not change the qualitative dependence on $m$, $T$, $(1 - 1/\delta)$, and $s_0 = \eta_0 LT$.

**Comparison to EF under PP.** Under partial participation with rate $p = m/K$, a standard diminishing-stepsize choice with $\sum_{r=0}^{R-1} \eta_r T = \Theta(\sqrt{R})$ yields an optimization term of order $\mathcal{O}(1/(p\sqrt{R}))$, equivalently $\mathcal{O}(\sqrt{K/m}/\sqrt{R})$. This matches the $\sqrt{K/m}$ slowdown under partial participation reported for EF-type methods (Li & Li, 2023, Theorem 4.10). The mini-batch noise term scales with $1/m$, whereas the compression- and heterogeneity-induced floor terms remain proportional to $(1 - 1/\delta)$ (and inherit the same $T$-dependence as in the full-participation bound), consistent with prior analyses.

**Relation to EF21.** EF21 and its extensions (Richtárik et al., 2021; Fatkhullin et al., 2025) obtain stronger guarantees (in particular, they can avoid an error floor) in a different regime: synchronized data-parallel optimization with $T$=1 (no local steps), where a *shared* iterate is updated and (stochastic) gradients or gradient differences are compressed at that common point. Our setting is complementary: federated local-SGD with $T > 1$ local steps (and possibly partial participation), where the compressed object is the *accumulated local update* and client drift plays a central role. Extending EF21-style arguments to this local-SGD setting with biased contractive compressors, or developing EF21-type step-ahead variants, appears non-trivial and is left for future work.

### 4.4 Uniform residual contraction: useful special cases

**Corollary 2** (Uniform residual contraction for $\alpha \equiv 1$). *Under the conditions of Lemma 4, assume $\eta_r \equiv \eta_0$ so that $s_r \equiv s_0 := \eta_0 LT \leq 1/8$ and choose $\alpha_r \equiv 1$. Then*

$$\rho_{\max} = \left(1 - \tfrac{1}{\delta}\right) 24 s_0^2 \ \leq \ 24 s_0^2 \ \leq \ \tfrac{3}{8}, \qquad hence \qquad \frac{1}{1 - \rho_{\max}} \ \leq \ \frac{8}{5}.$$

*Consequently, $\frac{\mathcal{E}_{\max}}{1 - \rho_{\max}} \leq \frac{8}{5} \mathcal{E}_{\max}$.*

**Corollary 3** (Uniform residual contraction for $\alpha \equiv \alpha^\star$). *Under the conditions of Lemma 4, assume $\eta_r \equiv \eta_0$ so that $s_r \equiv s_0 := \eta_0 LT \leq 1/8$ and choose $\alpha_r \equiv \alpha^\star := \frac{1}{1 + 12 s_0^2}$. Then*

$$\rho_{\max} = \left(1 - \tfrac{1}{\delta}\right) \frac{24 s_0^2}{1 + 12 s_0^2} \ \leq \ \frac{24 s_0^2}{1 + 12 s_0^2} \ \leq \ \frac{6}{19}, \qquad so \qquad \frac{1}{1 - \rho_{\max}} \leq \frac{19}{13}.$$

*Consequently, $\frac{\mathcal{E}_{\max}}{1 - \rho_{\max}} \leq \frac{19}{13} \mathcal{E}_{\max}$.*

**Practical takeaway.** When $s_r = \eta_r LT$ is small and $\alpha_r$ is chosen near $\alpha_r^\star$, the residual contraction is strictly stronger than in EF (cf. Prop. 1), leading to smaller constants in the residual-induced terms and typically a faster initial decrease in objective value and gradient norm for a fixed communication budget. In regimes with aggressive compression or pronounced heterogeneity, SA-PEF can therefore outperform both standard EF and uncompressed Local-SGD. To place SA-PEF in context, Table 2 contrasts Fed-EF, SAEF, CSER, SCAFCOM, and SA-PEF, highlighting that SA-PEF targets the same FL regime as Fed-EF while improving residual contraction under biased compression, whereas SCAFCOM relaxes heterogeneity assumptions at the cost of additional control-variates state.

## 5 Experiments

### 5.1 Experimental setup

We evaluate SA-PEF on three image classification benchmarks of increasing difficulty and scale. We use CIFAR-10 (Krizhevsky, 2009) with ResNet-9 (Page, 2024), CIFAR-100 (Krizhevsky, 2009) with ResNet-18 (He et al., 2016), and Tiny-ImageNet (Le & Yang, 2015) with ResNet-34 (He et al., 2016), trained with cross-entropy loss. We apply standard preprocessing: per-dataset mean/std normalization. We create $K = 100$

clients and adopt *partial participation* with rate $p \in \{0.1, 0.5, 1.0\}$ where, in each round $r$, the server samples $m = \lfloor pK \rfloor$ clients uniformly without replacement. To induce client data heterogeneity, we apply *Dirichlet* label partitioning with concentration parameter $\gamma \in \{0.1, 0.5, 1.0\}$, where smaller $\gamma$ indicates stronger non-IID (Hsu et al., 2019). Each client's local dataset remains fixed across rounds. Each selected client performs $T = 5$ local SGD steps per round. Training runs for $R = 200$ communication rounds, except for the CIFAR-10 $p = 0.1$ settings in Figures 1a and 3a, which are extended to $R = 400$ rounds. Unless otherwise stated, the local mini-batch size is 64, momentum is 0.9, and weight decay is $5 \times 10^{-4}$ on CIFAR and $1 \times 10^{-4}$ on Tiny-ImageNet. We use Top-$k$ sparsification with sparsity level $k/d \in \{0.01, 0.05, 0.1\}$. Clients transmit both indices and values of selected entries. We compare SA-PEF with uncompressed LocalSGD, EF (Li & Li, 2023), SAEF (Xu et al., 2021), and CSER (Xie et al., 2020). All methods use the same client sampling, optimizer, learning-rate schedules, and total communication budget (rounds and bits). We report Top-1 accuracy versus rounds and communicated bits, rounds-to-target accuracy, and final accuracy at a fixed communication budget. We repeat all experiments with five random seeds and report mean values in the plots. For CIFAR-10 and CIFAR-100, we additionally report the final test accuracy as mean $\pm$ standard deviation over these five runs in Table 3. Due to space constraints, we present results under high compression, with comprehensive results across all settings in the Appendix B.

## 5.2 Empirical results

Figure 1 compares SA-PEF with FedAvg (dense) and compressed baselines (EF, SAEF, CSER) on CIFAR-10 with ResNet-9 under two participation rates ($p \in \{0.1, 0.5\}$) and two compression budgets (Top-1%, Top-10%). In the accuracy versus rounds plots (top row), SA-PEF generally reaches a given accuracy in fewer rounds than EF and SAEF, with the largest margin in the harder regime ($\gamma$=0.1, Top-1%). SAEF often shows an initial jump but tends to plateau, while SA-PEF continues to improve. CSER typically lags early and only catches up later. In the accuracy versus communication plots (bottom row), SA-PEF's curves are left-shifted: for the same test accuracy it requires less uplink communication (in GB) than EF or SAEF, whereas FedAvg attains high accuracy only at orders of magnitude higher cost. Raising participation to $p$=0.5 benefits all approaches and narrows round-wise gaps, but SA-PEF remains the most communication-efficient across schemes.[1]

Figure 2 shows results on CIFAR-100 with ResNet-18. Despite the increased task difficulty, the same qualitative trends persist: SA-PEF tends to dominate early rounds and delivers higher accuracy per unit of communication across most regimes, SAEF often plateaus early, and CSER improves mainly at larger communication budgets.[2] The gains are most pronounced under aggressive compression (Top-1%) and low participation (e.g., $p$=0.1). Overall, these results suggest that combining preview with partial error feedback provides faster early progress and superior accuracy-communication trade-offs across architectures and datasets.

We also evaluate SA-PEF on FEMNIST (Caldas et al., 2018), a writer-partitioned federated benchmark. Full setup details and plots are provided in Appendix B.2.1. The FEMNIST results support two main conclusions: (i) on the natural writer partition, SA-PEF remains competitive with dense FedAvg and is the strongest compressed method among the baselines considered; and (ii) under a substantially more heterogeneous pathological-2 split on the same dataset, the advantage of SA-PEF over EF and SAEF becomes markedly more pronounced.

**Discussion.** Overall, our results position SA-PEF as a lightweight but effective upgrade of classical EF in FL. Compared to EF and its step-ahead variant SAEF, SA-PEF converges consistently faster and offers better accuracy-communication trade-offs under practical settings. Relative to CSER, which periodically resets residuals to control mismatch, SA-PEF achieves comparable or better robustness without introducing any additional reset-period hyperparameter, and it avoids CSER's high *peak* communication cost when

---

[1] Under low participation (e.g., 1-10%), effective batch sizes shrink and both drift and compression noise increase, hence participation-aware hyperparameters (e.g., learning rate, local steps $T$, or $\alpha_r$) may need tuning. Here, we fix hyperparameters across methods for fairness, which can reduce the observable advantage of SA-PEF in extreme low-participation regimes.

[2] As in CIFAR-10, under low participation (1–10%), the differences between methods may be less visible without participation-aware tuning of hyperparameters.

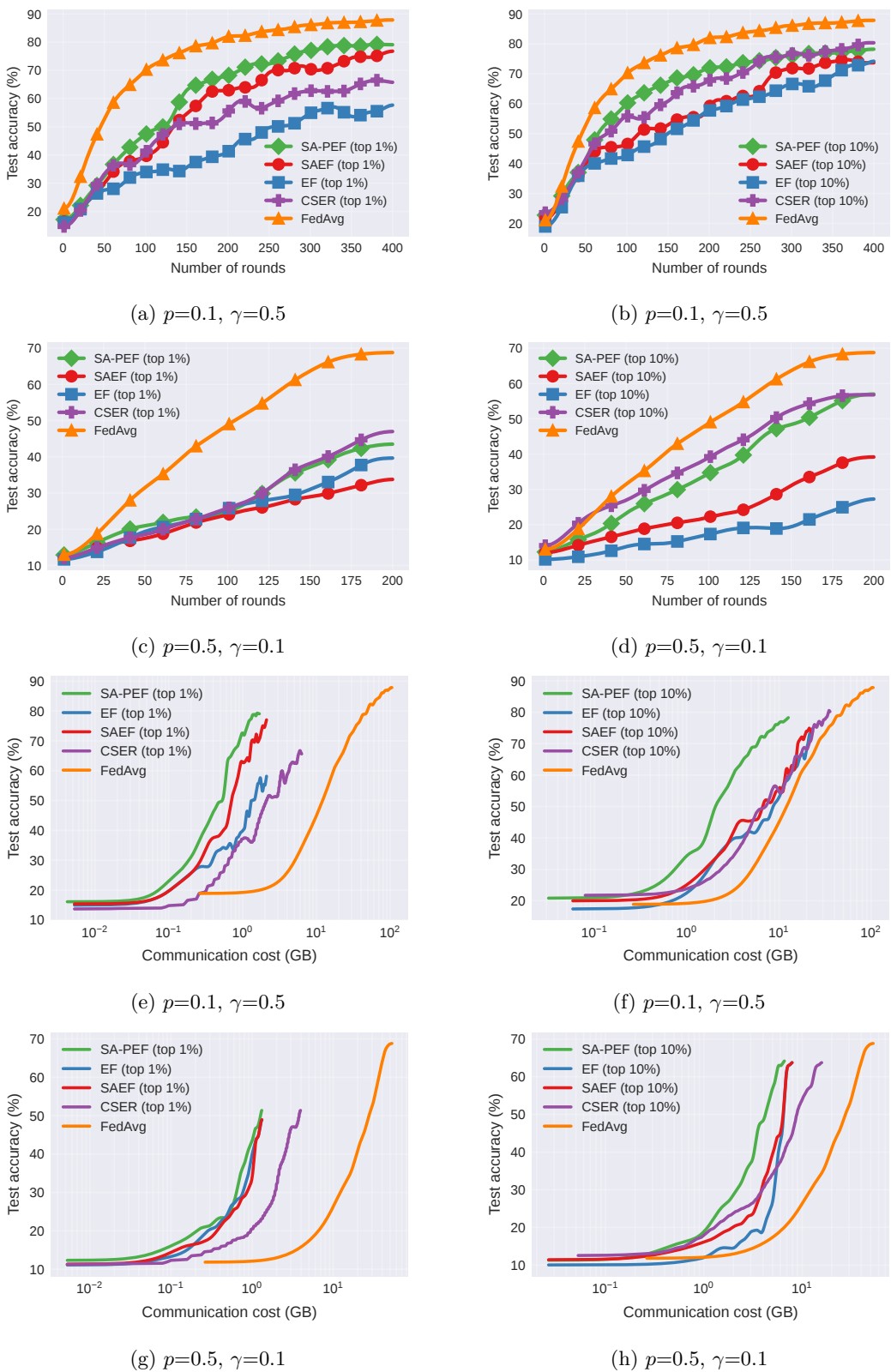

Figure 1: Test accuracy vs. number of rounds (rows 1–2) and communicated GB (rows 3–4) on the CIFAR-10 dataset using ResNet-9.

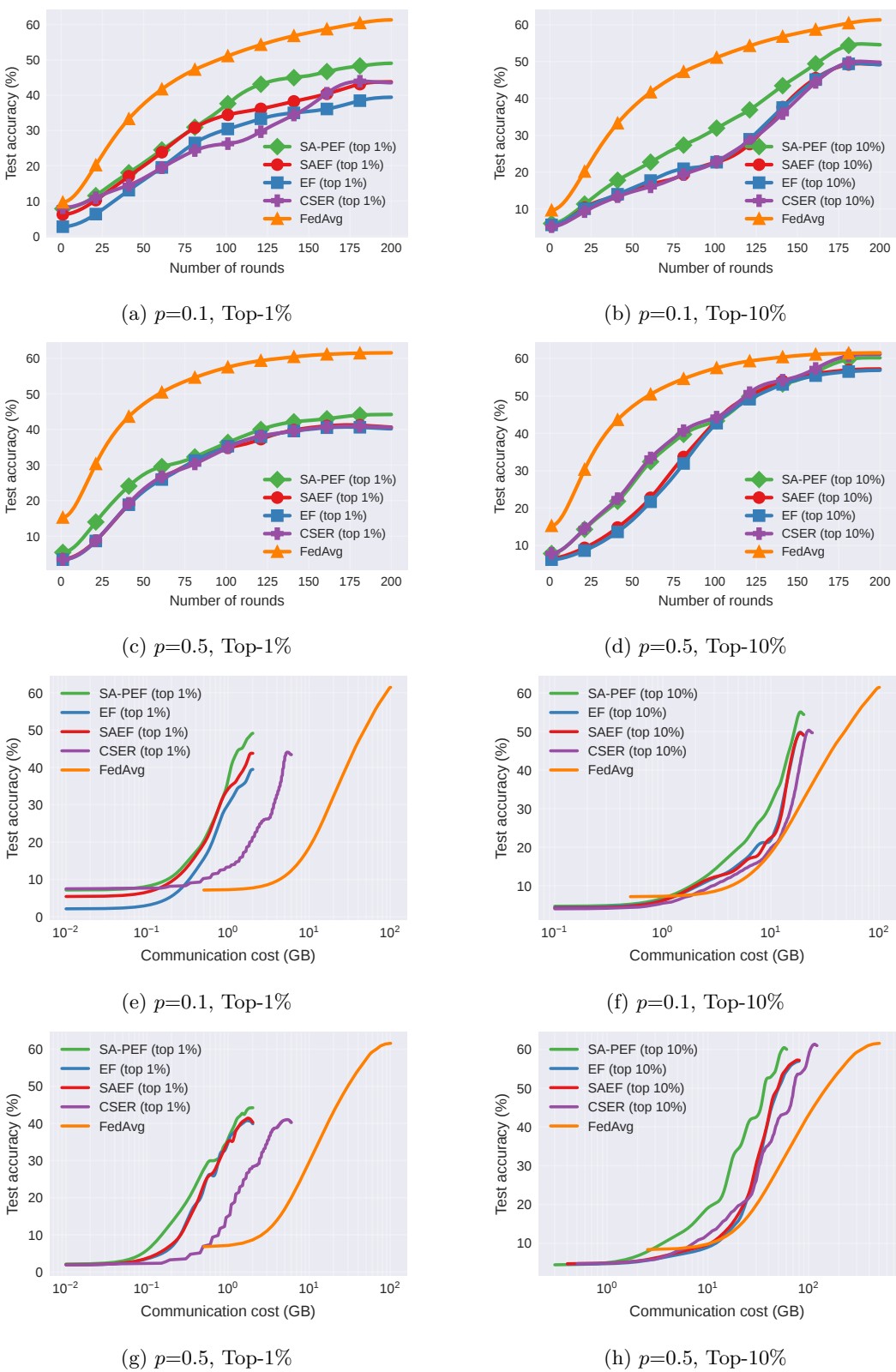

Figure 2: Test accuracy vs. number of rounds (rows 1–2) and communicated GB (rows 3–4) on the CIFAR-100 dataset using ResNet-18 and $\gamma$=0.1.

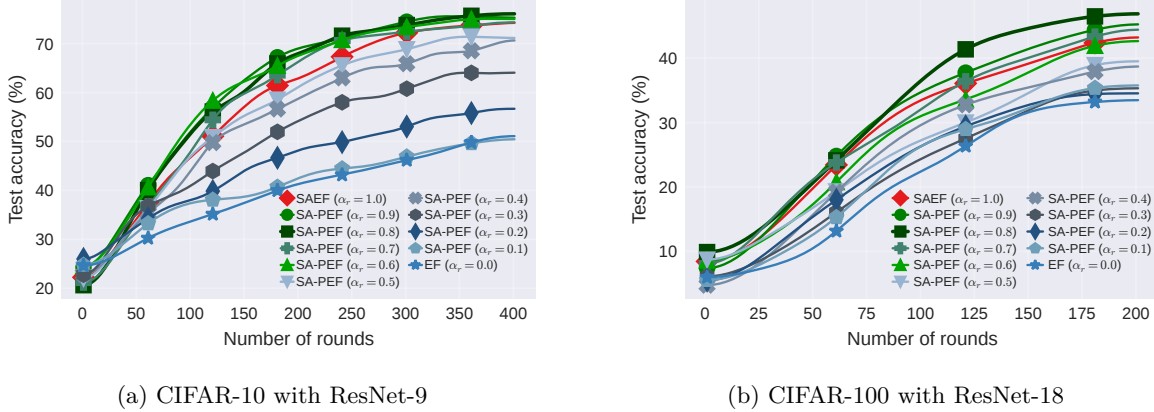

(a) CIFAR-10 with ResNet-9          (b) CIFAR-100 with ResNet-18

Figure 3: Sensitivity analysis of step-ahead coefficient $\alpha$.

compressed or full residuals are transmitted at reset rounds. Since practical systems must provision for peak, rather than average, bandwidth and latency, this makes SA-PEF more attractive as a drop-in component in resource-constrained deployments. Control-variate methods such as SCAFFOLD and SCAFCOM target a complementary axis, mitigating client drift and heterogeneity via additional per-client state, whereas SA-PEF focuses on reducing compression-induced residual mismatch within the EF family. In this sense, SA-PEF is best viewed as a drop-in improvement for EF-style compressed FL (and a natural building block for future combinations with control variates), providing significant gains in challenging regimes with high compression and strong heterogeneity.

### 5.3 Sensitivity analysis of step-ahead coefficient $\alpha$

To assess the robustness of SA-PEF to the choice of step-ahead coefficient, we sweep $\alpha_r$ between zero and one in increments of 0.1 on CIFAR-10 with ResNet-9 and CIFAR-100 with ResNet-18 under non-IID Dirichlet partitioning ($\gamma = 0.1$), Top-1% sparsification, $T = 5$ local steps, and $p = 0.1$ participation. Figure 3 reports test accuracy versus rounds for EF ($\alpha_r = 0$), SAEF ($\alpha_r = 1$), and SA-PEF with intermediate $\alpha_r$ values. Three regimes emerge: (i) *Small $\alpha$* ($\alpha_r \leq 0.3$) behaves similarly to EF, with noticeably slower convergence and lower final accuracy. (ii) *Intermediate $\alpha$* ($\alpha_r \in [0.6, 0.9]$) produces nearly identical curves, yielding the fastest convergence and highest final accuracy. This interval includes the default $\alpha_r = 0.85$ used in our main experiments. (iii) *Full step-ahead* ($\alpha_r = 1.0$, SAEF) accelerates early rounds but plateaus slightly below the best SA-PEF setting in the later phases. Overall, SA-PEF is robust across a broad high-$\alpha$ region, while performance is significantly affected only at the extremes: $\alpha_r \approx 0$ (reducing to EF) or $\alpha_r = 1$ (SAEF). This supports treating $\alpha$ as a momentum-like parameter and using a single default value (e.g., $\alpha_r \approx 0.8$–$0.9$) across tasks without heavy tuning. A natural extension is to adapt $\alpha_r$ online based on an estimate of the local-work scale $s_r$. Since our analysis identifies $\alpha_r^\star = 1/(1 + 12 s_r^2)$ as the minimizer of the contraction factor, one plausible choice is $\alpha_r = 1/(1 + 12 \hat{s}_r^2)$, where $\hat{s}_r$ is an online estimate of $s_r$. We leave this adaptive design to future work.

### 5.4 Gradient mismatch

Let $w \in \mathbb{R}^d$ stack all trainable parameters, and let $f(\cdot; w)$ be the network in *evaluation* mode (dropout disabled, BatchNorm with frozen statistics). For a mini-batch $S = \{(x_i, y_i)\}_{i=1}^b$ drawn from a held-out loader, define the batch loss as $\mathcal{L}_S(w) = \frac{1}{b} \sum_{(x,y) \in S} \ell(f(x; w), y)$. At round $r$, for client $k$ and $\alpha \in [0, 1]$, we consider two evaluation points $w_r$ and $w_r^{(\alpha,k)} = w_r - \alpha\, e_r^{(k)}$. Using the same mini-batch $S$ for both, we compute the associated gradients as $g_r = \nabla_w \mathcal{L}_S(w_r)$ and $g_r^{(\alpha,k)} = \nabla_w \mathcal{L}_S(w_r^{(\alpha,k)})$, and define the squared gradient-mismatch as $\hat{\varepsilon}_r^{(k)}(\alpha) = \|g_r - g_r^{(\alpha,k)}\|_2^2$ and its client average as $\hat{\varepsilon}_r(\alpha) = \frac{1}{K} \sum_{k=1}^K \hat{\varepsilon}_r^{(k)}(\alpha)$. In particular, we switch the model to `eval` mode, clear any stale gradients, reuse the same $S$, compute $g_r$ and $g_r^{(\alpha,k)}$ via

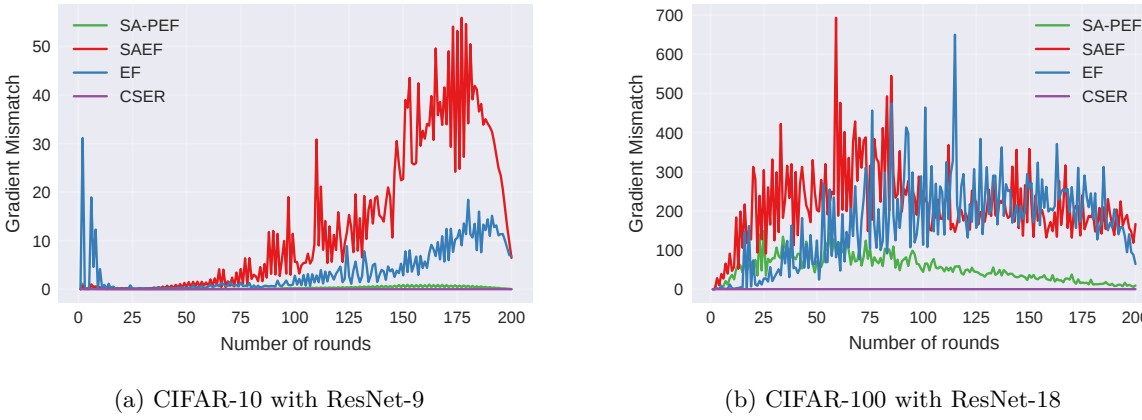

(a) CIFAR-10 with ResNet-9        (b) CIFAR-100 with ResNet-18

Figure 4: Gradient mismatch for different algorithms.

first-order autodiff, and then restore the model to training mode. We do not alter any optimizer state or BatchNorm buffers.

Figure 4 shows the gradient-mismatch probe across training rounds for all methods. SA-PEF keeps the mismatch essentially flat and near zero throughout training: previewing only a fraction of the residual shifts the evaluation point closer to where EF applies the update, while the retained $(1 - \alpha)e_r$ further prevents residual build-up. EF exhibits a steady late-phase rise, consistent with its one-step staleness effect as displacement accumulates over rounds. SAEF produces the largest spikes and highest overall mismatch, as with full preview ($\alpha{=}1$), the evaluation occurs at $w_r - e_r$. Large or heterogeneous residuals spike mismatch, causing late-stage plateaus. CSER stays near zero, reflecting its error-reset/averaging mechanism that suppresses persistent residual drift. Overall, the results confirm that combining partial preview with partial EF controls mismatch throughout training, unlike plain EF or SAEF.

## 6 Conclusion

We presented SA-PEF, which combines a step-ahead preview with partial error feedback to accelerate early-round progress in federated learning while preserving the long-run stability of EF under biased compression. Our analysis covers nonconvex objectives, local SGD with $T > 1$ steps, partial participation, and $\delta$-contractive compressors. It yields the standard $\mathcal{O}(1/R)$ stationarity behavior (up to constants) together with an $R$-independent variance/heterogeneity floor induced by biased compression. A key takeaway is that step-ahead reshapes the residual dynamics through a contraction factor $\rho_r$, explaining the empirically observed early-phase speedups and the benefit of choosing $\alpha_r$ near its theory-guided optimum to balance rapid warm-up against late-stage stability. Looking ahead, a promising direction is to develop *adaptive schedules* for $\{\alpha_r\}$ based on online estimates of residual norms, client drift, or noise levels, and to analyze the resulting feedback between mismatch reduction and residual evolution under partial participation. Another direction is integrating SA-PEF with *adaptive optimizers and momentum*, which calls for a preconditioned drift analysis to control momentum–residual interactions. Finally, it would be valuable to bring ideas from EF21 into the federated local-SGD regime studied here, e.g., by designing and analyzing EF21-style mechanisms that remain effective under $T > 1$ local steps, client subsampling, and biased contractive compression.

**Broader Impact**

SA-PEF addresses a practical bottleneck in federated learning by improving the stability of error-feedback mechanisms under biased compression and partial participation. By reducing communication per round, it can make deployments more feasible on bandwidth-limited networks and resource-constrained devices, potentially broadening participation and lowering communication-related energy use. At the same time, federated learning is often used in sensitive or high-stakes domains. Improved efficiency may lower the barrier to deployment, increasing the importance of privacy, security, and governance safeguards. SA-PEF does not

itself provide privacy or security guarantees, hence deployments should pair it with mechanisms such as secure aggregation or differential privacy, along with appropriate access control, auditing, and policy oversight. Client-level heterogeneity also raises fairness concerns that improved optimization alone does not resolve. Compression and client subsampling can disproportionately attenuate updates from clients with rare, minority, or outlier data, potentially widening performance gaps even when average accuracy improves. While SA-PEF reduces early-round stalling via tighter residual control, it does not guarantee fairness or uniform performance across clients or subgroups. We therefore encourage reporting beyond average metrics (e.g., worst-client or percentile accuracy, client-level variance, subgroup gaps, and participation-conditioned performance) and evaluating complementary mitigation strategies such as fairness-aware aggregation, adaptive compression, or $\alpha$-scheduling informed by client drift or disparity indicators.

## Acknowledgments

This work was supported in part by the Research Council of Norway through the Privacy@Edge project (Project No. 338909) and by the Research Council of Finland (Grant No. 354523). We also thank Aalto Science-IT for providing computational resources.

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

## A   Proof of Convergence Results

### A.1   Proof of Theorem 1

*Proof.* Subtracting the *average residual* from the communicated model converts a biased, compressed update into something that resembles vanilla SGD.

We have:

1. **Step-ahead shift** $w^{(k)}_{r+\frac{1}{2}} = w^{(k)}_r - \alpha_r e^{(k)}_r$.

2. **Residual update & compression** $e^{(k)}_{r+1} = u^{(k)}_{r+1} - C(u^{(k)}_{r+1})$ with $u^{(k)}_{r+1} = (1 - \alpha_r)e^{(k)}_r + g^{(k)}_r$.

Let

$$\bar{e}_r := \frac{1}{K}\sum_{k=1}^{K} e^{(k)}_r, \quad \bar{u}_{r+1} := (1 - \alpha_r)\bar{e}_r + \bar{g}_r, \quad \bar{C}_{r+1} := \frac{1}{K}\sum_k C(u^{(k)}_{r+1}),$$

so the server update is $w_{r+1} = w_r - \eta\,\bar{C}_{r+1}$.

Define a virtual iterate:

$$
\begin{aligned}
x_r &= w_r - \eta\,\bar{e}_r \\
x_{r+1} &= w_{r+1} - \eta\,\bar{e}_{r+1} \\
&= (w_r - \eta\,\bar{C}_{r+1}) - \eta\,(\bar{u}_{r+1} - \bar{C}_{r+1}) \\
&= w_r - \eta\,\bar{u}_{r+1} \\
&= (x_r + \eta\,\bar{e}_r) - \eta\,[(1 - \alpha_r)\bar{e}_r + \bar{g}_r] \\
&= x_r + \eta\,\alpha_r\bar{e}_r - \eta\,\bar{g}_r.
\end{aligned}
$$

$$f(x_{r+1}) \leq f(x_r) + \langle \nabla f(x_r),\, x_{r+1} - x_r \rangle + \frac{L}{2}\left\|x_{r+1} - x_r\right\|^2. \tag{5}$$

Because $x_{r+1} - x_r = \eta\big(\alpha_r\bar{e}_r - \bar{g}_r\big)$, taking the expectation over all randomness inside round $r$ yields

$$\mathbb{E}[f(x_{r+1})] \leq \mathbb{E}[f(x_r)] + \eta\,\mathbb{E}[\langle \nabla f(x_r),\, \alpha_r\bar{e}_r - \bar{g}_r \rangle] + \frac{\eta^2 L}{2}\,\mathbb{E}[\left\|\alpha_r\bar{e}_r - \bar{g}_r\right\|^2]. \tag{6}$$

Isolating the drift of the objective between two consecutive rounds and decomposing the inner product gives

$$\mathbb{E}[f(x_{r+1})] - \mathbb{E}[f(x_r)] \leq \eta \, \mathbb{E}[\langle \nabla f(x_r), \, \alpha_r \bar{e}_r - \bar{g}_r \rangle] + \frac{\eta^2 L}{2} \, \mathbb{E}[\|\alpha_r \bar{e}_r - \bar{g}_r\|^2]$$

$$= \underbrace{\eta \, \mathbb{E}[\langle \nabla f(w_r), \, \alpha_r \bar{e}_r - \bar{g}_r \rangle]}_{T1} + \underbrace{\frac{\eta^2 L}{2} \, \mathbb{E}[\|\alpha_r \bar{e}_r - \bar{g}_r\|^2]}_{T2}$$

$$+ \underbrace{\eta \, \mathbb{E}[\langle \nabla f(x_r) - \nabla f(w_r), \, \alpha_r \bar{e}_r - \bar{g}_r \rangle]}_{T3}. \tag{7}$$

**Bounding the first inner-product term**  Recall

$$T_1 = \eta \, \mathbb{E}_r[\langle \nabla f(w_r), \, \alpha_r \bar{e}_r - \bar{g}_r \rangle] = \eta \, \alpha_r \, \langle \nabla f(w_r), \bar{e}_r \rangle - \eta \, \mathbb{E}_r[\langle \nabla f(w_r), \bar{g}_r \rangle],$$

where $\bar{g}_r = \frac{\eta_r}{K} \sum_{k=1}^{K} \sum_{t=0}^{T-1} (\nabla f_k(w_{r+\frac{1}{2},t}^{(k)}) + \xi_{k,t})$ and $\mathbb{E}_r[\xi_{k,t}] = 0$.

Define $\bar{\nabla}_t := \frac{1}{K} \sum_{k=1}^{K} \nabla f_k(w_{r+\frac{1}{2},t}^{(k)})$ and $\Delta_t = \frac{1}{K} \sum_{k=1}^{K} (\nabla f_k(w_{r+\frac{1}{2},t}^{(k)}) - \nabla f(w_r))$

Write $\bar{\nabla}_t = \nabla f(w_r) + \Delta_t$ and use $L$-smoothness to get $\mathbb{E}_r \|\Delta_t\|^2 \leq L^2 \times \frac{1}{K} \sum_k \mathbb{E}_r \|w_{r+\frac{1}{2},t}^{(k)} - w_r\|^2$.

Young's inequality gives, for any $\lambda_1 > 0$,

$$\mathbb{E}_r \langle \nabla f(w_r), \bar{g}_r \rangle \geq \eta_r T \Big(1 - \frac{\lambda_1}{2}\Big) \|\nabla f(w_r)\|^2 - \frac{\eta_r L^2}{2\lambda_1} S, \quad S := \frac{1}{K} \sum_{k=1}^{K} \sum_{t=0}^{T-1} \mathbb{E}_r \|w_{r+\frac{1}{2},t}^{(k)} - w_r\|^2.$$

By Lemma 2 (summed over $t$),

$$S \leq 12 \, \eta_r^2 T^2 \sigma^2 + 168 \, \eta_r^2 T^3 \nu^2 + 36 \, \eta_r^2 T^3 \beta^2 \|\nabla f(w_r)\|^2 + 3T \alpha_r^2 \bar{E}_r.$$

For the step-ahead piece, Young's inequality yields, for any $\lambda_2 > 0$,

$$\eta \, \alpha_r \, \langle \nabla f(w_r), \bar{e}_r \rangle \leq \frac{\eta \lambda_2}{2} \|\nabla f(w_r)\|^2 + \frac{\eta \, \alpha_r^2}{2\lambda_2} \bar{E}_r.$$

Combining,

$$T_1 \leq \eta \Big( \frac{\lambda_2}{2} - \eta_r T \Big(1 - \frac{\lambda_1}{2}\Big) \Big) \|\nabla f(w_r)\|^2 + \frac{\eta \, \alpha_r^2}{2\lambda_2} \bar{E}_r + \frac{\eta \, \eta_r L^2}{2\lambda_1} S.$$

Choose $\lambda_1 = 1$ and $\lambda_2 = \eta_r T/2$; then

$$T_1 \leq -\frac{\eta \, \eta_r T}{4} \|\nabla f(w_r)\|^2 + \frac{\eta \, \alpha_r^2}{\eta_r T} \bar{E}_r + \frac{\eta \, \eta_r L^2}{2} S,$$

and substituting the bound on $S$ gives

$$T_1 \leq \eta \, \eta_r \Big[ -\frac{T}{4} + 18 \, \eta_r^2 L^2 \beta^2 T^3 \Big] \|\nabla f(w_r)\|^2$$

$$+ \eta \, \alpha_r^2 \Big[ \frac{1}{\eta_r T} + \frac{3}{2} \eta_r L^2 T \Big] \bar{E}_r + 6 \, \eta \, \eta_r^3 L^2 T^2 \, \sigma^2 + 84 \, \eta \, \eta_r^3 L^2 T^3 \, \nu^2. \tag{8}$$

**Bounding the quadratic (smoothness) term**

Recall the second contribution in the descent inequality

$$T_2 = \frac{L \eta^2}{2} \, \mathbb{E}_r \big[ \|\alpha_r \bar{e}_r - \bar{g}_r\|^2 \big]. \tag{8}$$

**Bounding the gradient–mismatch term**

$$T_3 := \eta \, \mathbb{E}_r\big[\langle \nabla f(x_r) - \nabla f(w_r), \, \alpha_r \bar{e}_r - \bar{g}_r \rangle\big].$$

Since $x_r = w_r - \eta \bar{e}_r$, we have $\|x_r - w_r\| = \eta \|\bar{e}_r\|$, and by $L$-smoothness,

$$\|\nabla f(x_r) - \nabla f(w_r)\| \leq L\|x_r - w_r\| = L\eta \|\bar{e}_r\|.$$

Apply Young's inequality (any $\lambda > 0$) with $a = \nabla f(x_r) - \nabla f(w_r)$ and $b = \alpha_r \bar{e}_r - \bar{g}_r$:

$$\big|\langle \nabla f(x_r) - \nabla f(w_r), \, \alpha_r \bar{e}_r - \bar{g}_r \rangle\big| \leq \frac{\lambda}{2} L^2 \eta^2 \|\bar{e}_r\|^2 + \frac{1}{2\lambda} \|\alpha_r \bar{e}_r - \bar{g}_r\|^2.$$

Taking $\mathbb{E}_r[\cdot]$, using $\|\bar{e}_r\|^2 \leq \bar{E}_r$, and multiplying by $\eta$ yields

$$|T_3| \leq \frac{\lambda L^2 \eta^3}{2} \bar{E}_r + \frac{\eta}{2\lambda} \mathbb{E}_r\big[\|\alpha_r \bar{e}_r - \bar{g}_r\|^2\big].$$

Choosing $\lambda = \dfrac{1}{L\eta}$ balances the two terms and $T_3 \leq |T_3|$ gives

$$T_3 \leq \frac{L\eta^2}{2} \bar{E}_r + \frac{L\eta^2}{2} \mathbb{E}_r\big[\|\alpha_r \bar{e}_r - \bar{g}_r\|^2\big]. \tag{9}$$

Put the pieces together:

$$
\begin{aligned}
\mathbb{E}_r\big[f(x_{r+1}) - f(x_r)\big] \leq{}& \eta \, \eta_r \Big[-\frac{T}{4} + 18 \, \eta_r^2 L^2 \beta^2 T^3\Big] \|\nabla f(w_r)\|^2 \\
&+ \eta \, \alpha_r^2 \Big[\frac{1}{\eta_r T} + \frac{3}{2} \eta_r L^2 T\Big] \bar{E}_r + 6 \, \eta \, \eta_r^3 L^2 T^2 \, \sigma^2 + 84 \, \eta \, \eta_r^3 L^2 T^3 \, \nu^2 \\
&+ L\eta^2 \, \mathbb{E}_r\big[\|\alpha_r \bar{e}_r - \bar{g}_r\|^2\big] + \frac{L\eta^2}{2} \bar{E}_r
\end{aligned}
\tag{10}
$$

Using Lemma 3,

$$
\begin{aligned}
\mathbb{E}_r\big[\|\alpha_r \bar{e}_r - \bar{g}_r\|^2\big] \leq{}& 2\alpha_r^2 \bar{E}_r + 8\eta_r^2 T^2 \|\nabla f(w_r)\|^2 \big(1 + 36 L^2 \eta_r^2 T^2 \beta^2\big) \\
&+ \frac{4\eta_r^2 T}{K} \sigma^2 + 96 L^2 \eta_r^4 T^3 \big(\sigma^2 + 14 T \nu^2\big) + 24 \alpha_r^2 \eta_r^2 L^2 T^2 \bar{E}_r.
\end{aligned}
$$

$$
\begin{aligned}
\mathbb{E}_r\big[f(x_{r+1}) - f(x_r)\big] \leq{}& \Big[\eta \, \eta_r\Big(-\tfrac{T}{4} + 18 \, \eta_r^2 L^2 \beta^2 T^3\Big) + L\eta^2\Big(8\eta_r^2 T^2 + 288 \, L^2 \eta_r^4 T^4 \beta^2\Big)\Big] \|\nabla f(w_r)\|^2 \\
&+ \Big[\eta \, \alpha_r^2\Big(\tfrac{1}{\eta_r T} + \tfrac{3}{2} \eta_r L^2 T\Big) + L\eta^2\Big(2\alpha_r^2 + 24 \, \alpha_r^2 \eta_r^2 L^2 T^2\Big) + \tfrac{L\eta^2}{2}\Big] \bar{E}_r, \\
&+ \Big[6 \, \eta \, \eta_r^3 L^2 T^2 + \tfrac{4L\eta^2 \eta_r^2 T}{K} + 96 \, L^3 \eta^2 \eta_r^4 T^3\Big] \sigma^2 \\
&+ \Big[84 \, \eta \, \eta_r^3 L^2 T^3 + 1344 \, L^3 \eta^2 \eta_r^4 T^4\Big] \nu^2
\end{aligned}
\tag{11}
$$

**Telescoping.** Summing over $r = 0, \ldots, R-1$ and taking total expectation,

$$\mathbb{E}\big[f(x_R) - f(x_0)\big] \leq \sum_{r=0}^{R-1} \Big\{A_r \, \mathbb{E}\|\nabla f(w_r)\|^2 + \mathcal{E}_r \, \mathbb{E}[\bar{E}_r] + V_r \, \sigma^2 + H_r \, \nu^2\Big\}, \tag{12}$$

where

$$A_r := \eta\,\eta_r\left(-\tfrac{T}{4} + 18\,\eta_r^2 L^2\beta^2 T^3\right) + L\eta^2\left(8\eta_r^2 T^2 + 288\,L^2\eta_r^4 T^4\beta^2\right),$$

$$\mathcal{E}_r := \eta\,\alpha_r^2\left(\tfrac{1}{\eta_r T} + \tfrac{3}{2}\eta_r L^2 T\right) + L\eta^2\left(2\alpha_r^2 + 24\,\alpha_r^2\eta_r^2 L^2 T^2\right) + \tfrac{L\eta^2}{2},$$

$$V_r := 6\,\eta\,\eta_r^3 L^2 T^2 + \tfrac{4L\eta^2\eta_r^2 T}{K} + 96\,L^3\eta^2\eta_r^4 T^3,$$

$$H_r := 84\,\eta\,\eta_r^3 L^2 T^3 + 1344\,L^3\eta^2\eta_r^4 T^4.$$

We now derive the main convergence guarantee. The analysis begins with the telescoped recursion from equation 12, which provides an upper bound on the function value progress, $\mathbb{E}[f(x_R) - f(x_0)]$. The key step is to ensure that the coefficient $A_r$ of the squared gradient norm is negative, which guarantees descent. The following lemma establishes sufficient conditions for this.

**Lemma 1** (Sufficient Conditions for Descent). *Let $s_r := \eta_r LT$. Assume that for all rounds $r$, the parameters satisfy $s_r \leq 1/8$ and the following two conditions:*

$$18\,\beta^2\,s_r^2 \leq \frac{1}{8}, \tag{13}$$

$$\eta \leq \frac{1}{256\,\beta^2\,L\,\eta_r T}, \tag{14}$$

*where it is assumed that $\beta^2 \geq 1$. Then, the coefficient $A_r$ is bounded as*

$$A_r \leq -\frac{1}{16}\,\eta\,\eta_r T.$$

*Proof.* The coefficient $A_r$ is defined as

$$A_r = -\frac{\eta\eta_r T}{4} + 18\eta\eta_r^3 L^2\beta^2 T^3 + L\eta^2(8\eta_r^2 T^2 + 288L^2\eta_r^4 T^4\beta^2).$$

We proceed by bounding the two positive terms separately.

First, we bound the term $18\eta\eta_r^3 L^2\beta^2 T^3$. By applying the definition $s_r = \eta_r LT$ and condition equation 13, we obtain

$$18\eta\eta_r^3 L^2\beta^2 T^3 = (\eta\eta_r T) \times (18\beta^2\eta_r^2 L^2 T^2)$$
$$= (\eta\eta_r T) \times (18\beta^2 s_r^2) \leq \frac{1}{8}\eta\eta_r T.$$

Next, we bound the term $L\eta^2(8\eta_r^2 T^2 + 288L^2\eta_r^4 T^4\beta^2)$. Condition equation 13 implies $36\beta^2 s_r^2 \leq 1/4$. This allows us to simplify the expression within parentheses:

$$8\eta_r^2 T^2 + 288L^2\eta_r^4 T^4\beta^2 = 8\eta_r^2 T^2\left(1 + 36\beta^2 L^2\eta_r^2 T^2\right)$$
$$= 8\eta_r^2 T^2(1 + 36\beta^2 s_r^2)$$
$$\leq 8\eta_r^2 T^2\left(1 + \frac{1}{4}\right) = 10\eta_r^2 T^2.$$

Using this intermediate result and condition equation 14, we bound the full term. The assumption $\beta^2 \geq 1$ ensures that $256\beta^2 \geq 160$.

$$L\eta^2(8\eta_r^2 T^2 + 288L^2\eta_r^4 T^4\beta^2) \leq L\eta^2(10\eta_r^2 T^2)$$
$$= (10L\eta\eta_r^2 T^2) \times \eta$$
$$\leq 10L\eta\eta_r^2 T^2 \times \left(\frac{1}{256\beta^2 L\eta_r T}\right)$$
$$= \frac{10}{256\beta^2}\eta\eta_r T \leq \frac{10}{160}\eta\eta_r T = \frac{1}{16}\eta\eta_r T.$$

Finally, substituting these bounds back into the expression for $A_r$ yields the desired result:

$$A_r \leq -\frac{\eta\eta_r T}{4} + \frac{1}{8}\eta\eta_r T + \frac{1}{16}\eta\eta_r T$$
$$= \left(-\frac{4}{16} + \frac{2}{16} + \frac{1}{16}\right)\eta\eta_r T$$
$$= -\frac{1}{16}\eta\eta_r T.$$

This completes the proof. $\qquad\square$

### A.1.1 General Convergence Rate for Decaying Step-Sizes

Applying the bound on $A_r$ from Lemma 1 to the main inequality equation 12 yields:

$$\mathbb{E}[f(x_R) - f(x_0)] \leq \sum_{r=0}^{R-1}\left(-\frac{\eta\eta_r T}{16}\mathbb{E}\|\nabla f(w_r)\|^2 + \mathcal{E}_r\,\mathbb{E}[\bar{E}_r] + V_r\,\sigma^2 + H_r\,\nu^2\right).$$

Rearranging the terms to isolate the sum of squared gradients and assuming the function is bounded below by $f_\star := \inf_x f(x)$, we obtain a bound on the weighted sum:

$$\sum_{r=0}^{R-1}\eta_r T\,\mathbb{E}\|\nabla f(w_r)\|^2 \leq \frac{16}{\eta}(f(x_0) - f_\star) + \frac{16}{\eta}\sum_{r=0}^{R-1}\mathbb{E}\big[E_r\,\bar{E}_r + V_r\,\sigma^2 + H_r\,\nu^2\big]. \tag{15}$$

The main challenge is to bound the term involving the residual energy, $\sum_r \mathcal{E}_r\mathbb{E}[\bar{E}_r]$. We first decouple the coefficient using the supremum $\mathcal{E}_{\max} := \sup_r \mathcal{E}_r$. Then, we apply the bound on the sum of residuals from Lemma 4, which is derived by unrolling the per-round recursion for $\bar{E}_r$:

$$\frac{16}{\eta}\sum_{r=0}^{R-1}\mathcal{E}_r\mathbb{E}[\bar{E}_r] \leq \frac{16\mathcal{E}_{\max}}{\eta}\sum_{r=0}^{R-1}\mathbb{E}[\bar{E}_r]$$
$$\leq \frac{16\mathcal{E}_{\max}}{\eta(1-\rho_{\max})}\bar{E}_0 + \frac{16\mathcal{E}_{\max}(1-1/\delta)}{\eta(1-\rho_{\max})}\sum_{r=0}^{R-1}\mathbb{E}[B_r^{(\nabla)} + B_r^{(\nu,\sigma)}]$$
$$\overset{(a)}{\leq} \frac{16\mathcal{E}_{\max}(1-1/\delta)}{\eta(1-\rho_{\max})}\sum_{r=0}^{R-1}\mathbb{E}[B_r^{(\nabla)} + B_r^{(\nu,\sigma)}]$$

where (a) uses $\bar{E}_0 = 0$ since $e_0^{(k)} \equiv 0$.

Substituting the residual recursion (Lemma 4) into equation 15 introduces forcing terms. Since $B_r^{(\nabla)}$ is proportional to $\|\nabla f(w_r)\|^2$, it produces a contribution of the form $\Theta\sum_{r=0}^{R-1}\eta_r T\,\mathbb{E}\|\nabla f(w_r)\|^2$, where we *define*

$$\Theta := \frac{16}{\eta} \times \frac{\mathcal{E}_{\max}}{1-\rho_{\max}}\left(1 - \frac{1}{\delta}\right)\beta^2 \sup_{0 \leq r < R}\frac{8\,\eta_r^2 T^2 + 288\,L^2\eta_r^4 T^4}{\eta_r T}$$
$$= \frac{16}{\eta} \times \frac{\mathcal{E}_{\max}}{1-\rho_{\max}}\left(1 - \frac{1}{\delta}\right)\beta^2 \sup_r\big(8\,\eta_r T + 288\,L^2\eta_r^3 T^3\big).$$

With this notation, the weighted inequality becomes

$$(1-\Theta)\sum_{r=0}^{R-1}\eta_r T\,\mathbb{E}\|\nabla f(w_r)\|^2$$
$$\leq \frac{16}{\eta}\left[(f(x_0) - f_\star) + \frac{\mathcal{E}_{\max}}{1-\rho_{\max}}\left(1 - \frac{1}{\delta}\right)\sum_{r=0}^{R-1}\mathbb{E}B_r^{(\nu,\sigma)} + \sum_{r=0}^{R-1}V_r\,\sigma^2 + \sum_{r=0}^{R-1}H_r\,\nu^2\right]. \tag{16}$$

We now assume a small-steps/compression regime that ensures $\Theta \leq \frac{1}{2}$. We can therefore move this term to the left-hand side and absorb it, which tightens the overall bound at the cost of multiplying the remaining terms on the right-hand side by a factor of 2.

To obtain a bound on the conventional average, compare it to the weighted sum with weights $q_r := \eta_r T > 0$. Let $q_{\min} := \min_{0 \leq r < R} q_r$, $S_R := \sum_{r=0}^{R-1} q_r$, and $\phi_R := \dfrac{S_R}{R\, q_{\min}} \geq 1$. Then, for nonnegative terms,

$$\frac{1}{R} \sum_{r=0}^{R-1} \mathbb{E}\|\nabla f(w_r)\|^2 \leq \frac{S_R}{R\, w_{\min}} \times \frac{1}{S_R} \sum_{r=0}^{R-1} w_r\, \mathbb{E}\|\nabla f(w_r)\|^2 \tag{17}$$

$$= \phi_R \times \frac{1}{S_R} \sum_{r=0}^{R-1} \eta_r T\, \mathbb{E}\|\nabla f(w_r)\|^2. \tag{18}$$

Combining this with the absorbed bound and noting $x_0 = w_0$ since $e_0^{(k)} \equiv 0$, we obtain

$$\frac{1}{R} \sum_{r=0}^{R-1} \mathbb{E}\|\nabla f(w_r)\|^2 \leq \ \phi_R\, \frac{32}{\eta\, S_R} \Bigg[ (f(w_0) - f_\star) + \sum_{r=0}^{R-1} V_r\, \sigma^2 + \sum_{r=0}^{R-1} H_r\, \nu^2$$

$$+ \ \frac{\mathcal{E}_{\max}}{1 - \rho_{\max}} \Big(1 - \tfrac{1}{\delta}\Big) \sum_{r=0}^{R-1} \Big(4\, \eta_r^2 T + 96\, L^2 \eta_r^4 T^3\Big)\, \sigma^2$$

$$+ \ \frac{\mathcal{E}_{\max}}{1 - \rho_{\max}} \Big(1 - \tfrac{1}{\delta}\Big) \sum_{r=0}^{R-1} \Big(8\, \eta_r^2 T^2 + 1344\, L^2 \eta_r^4 T^4\Big)\, \nu^2 \Bigg].$$

### A.1.2 Convergence Rate for a Constant Step-Size

In the simpler setting where the inner learning rate is constant, $\eta_r \equiv \eta_0$ for all $r$, the sum of weights is $S_R = R\eta_0 T$. Under the descent and absorption conditions, we obtain

$$\frac{1}{R} \sum_{r=0}^{R-1} \mathbb{E}\|\nabla f(w_r)\|^2 \ \leq \ \frac{32\,(f(w_0) - f^\star)}{\eta\, \eta_0 T\, R} \ + \ \Big(1 - \tfrac{1}{\delta}\Big)\Big[C_\sigma\, \eta_0^2 L^2 T\, \sigma^2 + C_\nu\, \eta_0^2 L^2 T^2\, \nu^2\Big]$$

$$+ \ \frac{128\, L\, \eta_0}{K}\, \sigma^2$$

with

$$C_\sigma = \frac{32}{\eta}\Big[6\, \eta\, \eta_0^2 L^2 T + 96\, L^3 \eta\, \eta_0^3 T^2\Big] + \frac{32}{\eta} \times \frac{\mathcal{E}_{\max}}{1 - \rho_{\max}}\Big[4\, \eta_0 + 96\, L^2 \eta_0^3 T^2\Big],$$

$$C_\nu = \frac{32}{\eta}\Big[84\, \eta\, \eta_0^2 L^2 T^2 + 1344\, L^3 \eta\, \eta_0^3 T^3\Big] + \frac{32}{\eta} \times \frac{\mathcal{E}_{\max}}{1 - \rho_{\max}}\Big[8\, \eta_0 T + 1344\, L^2 \eta_0^3 T^3\Big].$$

which proves the theorem. $\qquad\square$

### A.2 Lemmas

**Lemma 2** (Local-model drift under SA-PEF). *Fix a communication round $r$ and an inner-loop horizon $T \geq 1$. Assume $0 < \eta_r \leq \frac{1}{8LT}$ and Assumptions 1–3. Let $\bar{E}_r := \frac{1}{K} \sum_{k=1}^{K} \|e_r^{(k)}\|^2$. Then, for every local step $t \in \{0, \dots, T-1\}$,*

$$\frac{1}{K} \sum_{k=1}^{K} \mathbb{E}_r\Big[\|w_{r+\frac{1}{2},t}^{(k)} - w_r\|^2\Big] \leq 12\, \eta_r^2 T\, \sigma^2 + 168\, \eta_r^2 T^2\, \nu^2 + 36\, \eta_r^2 T^2\, \beta^2\, \|\nabla f(w_r)\|^2 + 3\, \alpha_r^2\, \bar{E}_r. \tag{19}$$

*Consequently, summing equation 19 over $t = 0, \dots, T-1$ gives*

$$\frac{1}{K} \sum_{k=1}^{K} \sum_{t=0}^{T-1} \mathbb{E}_r\|w_{r+\frac{1}{2},t}^{(k)} - w_r\|^2 \leq 12\, \eta_r^2 T^2 \sigma^2 + 168\, \eta_r^2 T^3 \nu^2 + 36\, \eta_r^2 T^3 \beta^2 \|\nabla f(w_r)\|^2 + 3T\alpha_r^2 \bar{E}_r.$$

*Proof.* All expectations are conditional on the randomness within round $r$. Define $u_{k,t} := w^{(k)}_{r+\frac{1}{2},t} - w_r$, so $u_{k,0} = -\alpha_r e_r^{(k)}$. The local update implies

$$u_{k,t+1} = u_{k,t} - \eta_r\big(\nabla f_k(w^{(k)}_{r+\frac{1}{2},t}) + \xi_{k,t}\big), \qquad \mathbb{E}_r[\xi_{k,t}] = 0, \quad \mathbb{E}_r\|\xi_{k,t}\|^2 \le \sigma^2.$$

Expanding the square and taking $\mathbb{E}_r[\cdot]$ yields

$$\mathbb{E}_r\|u_{k,t+1}\|^2 = \mathbb{E}_r\|u_{k,t}\|^2 - 2\eta_r\,\mathbb{E}_r\langle u_{k,t}, \nabla f_k(w^{(k)}_{r+\frac{1}{2},t})\rangle + \eta_r^2\,\mathbb{E}_r\|\nabla f_k(w^{(k)}_{r+\frac{1}{2},t})\|^2 + \eta_r^2\,\mathbb{E}_r\|\xi_{k,t}\|^2$$

$$\le \mathbb{E}_r\|u_{k,t}\|^2 - 2\eta_r\,\mathbb{E}_r\langle u_{k,t}, \nabla f_k(w^{(k)}_{r+\frac{1}{2},t})\rangle + \eta_r^2\,\mathbb{E}_r\|\nabla f_k(w^{(k)}_{r+\frac{1}{2},t})\|^2 + \eta_r^2\sigma^2.$$

Write $\nabla f_k(w^{(k)}_{r+\frac{1}{2},t}) = \nabla f_k(w_r) + \Delta_{k,t}$ with $\|\Delta_{k,t}\| \le L\|u_{k,t}\|$ (Assumption 1). Then

$$-2\eta_r\langle u_{k,t}, \Delta_{k,t}\rangle \le 2\eta_r L\|u_{k,t}\|^2.$$

Also, by Young's inequality with $\gamma = \frac{1}{4T}$,

$$-2\eta_r\langle u_{k,t}, \nabla f_k(w_r)\rangle \le \frac{1}{4T}\|u_{k,t}\|^2 + 4\eta_r^2 T\|\nabla f_k(w_r)\|^2.$$

Using $\eta_r \le \frac{1}{8LT}$ gives $2\eta_r L \le \frac{1}{4T}$, hence

$$-2\eta_r\,\mathbb{E}_r\langle u_{k,t}, \nabla f_k(w^{(k)}_{r+\frac{1}{2},t})\rangle \le \frac{1}{2T}\,\mathbb{E}_r\|u_{k,t}\|^2 + 4\eta_r^2 T\|\nabla f_k(w_r)\|^2.$$

Moreover,

$$\mathbb{E}_r\|\nabla f_k(w^{(k)}_{r+\frac{1}{2},t})\|^2 \le 2\|\nabla f_k(w_r)\|^2 + 2L^2\,\mathbb{E}_r\|u_{k,t}\|^2.$$

Combining these bounds yields

$$\mathbb{E}_r\|u_{k,t+1}\|^2 \le \Big(1 + \tfrac{1}{2T} + 2\eta_r^2 L^2\Big)\mathbb{E}_r\|u_{k,t}\|^2 + (2 + 4T)\eta_r^2\|\nabla f_k(w_r)\|^2 + \eta_r^2\sigma^2.$$

Averaging over $k$ and defining $\bar{D}_t := \frac{1}{K}\sum_{k=1}^{K}\mathbb{E}_r\|u_{k,t}\|^2$, Assumption 3 gives

$$\bar{D}_{t+1} \le A\,\bar{D}_t + (2 + 4T)\eta_r^2\big(\beta^2\|\nabla f(w_r)\|^2 + \nu^2\big) + \eta_r^2\sigma^2, \qquad A := 1 + \tfrac{1}{2T} + 2\eta_r^2 L^2.$$

Since $\eta_r \le \frac{1}{8LT}$, we have $2\eta_r^2 L^2 \le \frac{1}{32T^2}$, hence $A \le 1 + \frac{1}{T} \le e^{1/T}$ and for $t \le T$, $A^t \le e \le 3$. Unrolling the recursion gives

$$\bar{D}_t \le 3\bar{D}_0 + 6T\Big((2 + 4T)\eta_r^2(\beta^2\|\nabla f(w_r)\|^2 + \nu^2) + \eta_r^2\sigma^2\Big).$$

Finally, $\bar{D}_0 = \frac{1}{K}\sum_k\|u_{k,0}\|^2 = \alpha_r^2\bar{E}_r$, and loosening constants using $12T + 24T^2 \le 36T^2$ and $12T + 24T^2 \le 168T^2$ for $T \ge 1$ yields equation 19. □

**Lemma 3** (Second moment of the shifted average update)**.** *Let $K$ be the number of clients, $T \ge 1$ the number of local steps, $\eta_r \in (0, \frac{1}{8LT}]$ the local stepsize in round $r$, and $\alpha_r \in [0,1]$ the step-ahead parameter. Define*

$$\bar{e}_r = \frac{1}{K}\sum_{k=1}^{K}e_r^{(k)}, \qquad g_r^{(k)} = \eta_r\sum_{t=0}^{T-1}\big(\nabla f_k(w^{(k)}_{r+\frac{1}{2},t}) + \xi_{k,t}\big), \qquad \bar{g}_r = \frac{1}{K}\sum_{k=1}^{K}g_r^{(k)}.$$

*Under Assumptions 1–3,*

$$\mathbb{E}_r\big[\|\alpha_r\bar{e}_r - \bar{g}_r\|^2\big] \le 2\alpha_r^2\bar{E}_r + 8\eta_r^2 T^2\|\nabla f(w_r)\|^2\big(1 + 36L^2\eta_r^2 T^2\beta^2\big)$$

$$+ \frac{4\eta_r^2 T\sigma^2}{K} + 96L^2\eta_r^4 T^3\big(\sigma^2 + 14T\nu^2\big) + 24\alpha_r^2\eta_r^2 L^2 T^2\bar{E}_r. \tag{20}$$

*Here $\bar{E}_r = \dfrac{1}{K}\displaystyle\sum_{k=1}^{K}\|e_r^{(k)}\|^2$ is the average residual energy at the beginning of round $r$, and $w_r$ is the global model before local computation.*

*Proof.* Using Young's inequality, for any $a, b \in \mathbb{R}^d$, $\|a - b\|^2 \leq 2\|a\|^2 + 2\|b\|^2$. With $a = \alpha_r \bar{e}_r$ (deterministic given $\mathcal{F}_r$) and $b = \bar{g}_r$,

$$\|\alpha_r \bar{e}_r - \bar{g}_r\|^2 \leq 2\alpha_r^2 \|\bar{e}_r\|^2 + 2\|\bar{g}_r\|^2. \tag{21}$$

Since $\|\bar{e}_r\|^2 \leq \bar{E}_r$ (Jensen), the first term in equation 21 gives $2\alpha_r^2 \bar{E}_r$ after expectation.

**Bounding the second moment of $\bar{g}_r$ and the gradient–noise cross term.** Write $\bar{g}_r = \eta_r \sum_{t=0}^{T-1}(\bar{\nabla}_t + \bar{\xi}_t)$ with $\bar{\nabla}_t = \frac{1}{K}\sum_k \nabla f_k(w_{r+\frac{1}{2},t}^{(k)})$ and $\bar{\xi}_t = \frac{1}{K}\sum_k \xi_{k,t}$. By Young's inequality,

$$\mathbb{E}_r \|\bar{g}_r\|^2 \;\leq\; 2\eta_r^2 \, \mathbb{E}_r \Big\| \sum_{t=0}^{T-1} \bar{\nabla}_t \Big\|^2 \;+\; 2\eta_r^2 \, \mathbb{E}_r \Big\| \sum_{t=0}^{T-1} \bar{\xi}_t \Big\|^2.$$

Using independence across $(k,t)$ and Assumption 2, $\mathbb{E}_r\|\sum_t \bar{\xi}_t\|^2 = \sum_t \mathbb{E}_r\|\bar{\xi}_t\|^2 \leq T\sigma^2/K$.

**Bounding the summed local gradients.** Decompose $\nabla f_k(w_{r+\frac{1}{2},t}^{(k)}) = \nabla f_k(w_r) + \Delta_{k,t}$ with $\|\Delta_{k,t}\| \leq L\|w_{r+\frac{1}{2},t}^{(k)} - w_r\|$ (Assumption 1). Then

$$\frac{1}{K^2}\mathbb{E}_r \Big\| \sum_{k,t} \nabla f_k(w_{r+\frac{1}{2},t}^{(k)}) \Big\|^2 \leq 2T^2 \|\nabla f(w_r)\|^2 + \frac{2L^2 T}{K}\sum_{k,t} \mathbb{E}_r \|w_{r+\frac{1}{2},t}^{(k)} - w_r\|^2.$$

Insert the local-model drift (Lemma 2) and sum over $t$. For each $t \in \{0,\ldots,T-1\}$, Lemma 2 yields

$$\frac{1}{K}\sum_k \mathbb{E}_r \|w_{r+\frac{1}{2},t}^{(k)} - w_r\|^2 \leq 12\,\eta_r^2 T\,\sigma^2 + 168\,\eta_r^2 T^2\,\nu^2 + 36\,\eta_r^2 T^2\,\beta^2\,\|\nabla f(w_r)\|^2 + 3\alpha_r^2 \bar{E}_r.$$

Summing over $t = 0,\ldots,T-1$ gives

$$\frac{1}{K}\sum_{k,t} \mathbb{E}_r \|w_{r+\frac{1}{2},t}^{(k)} - w_r\|^2 \leq 12\,\eta_r^2 T^2\,\sigma^2 + 168\,\eta_r^2 T^3\,\nu^2 + 36\,\eta_r^2 T^3\,\beta^2\,\|\nabla f(w_r)\|^2 + 3T\alpha_r^2 \bar{E}_r.$$

Assemble,

$$\mathbb{E}_r \|\bar{g}_r\|^2 \leq 2\eta_r^2 \Big[ 2T^2 \|\nabla f(w_r)\|^2 + \frac{2L^2 T}{K}\sum_{k,t} \mathbb{E}_r \|w_{r+\frac{1}{2},t}^{(k)} - w_r\|^2 \Big] + 2\eta_r^2 \times \frac{T\sigma^2}{K}$$

$$\leq 4\eta_r^2 T^2 \|\nabla f(w_r)\|^2 \big(1 + 36L^2\eta_r^2 T^2\beta^2\big) + \frac{2\eta_r^2 T\sigma^2}{K} + 48L^2\eta_r^4 T^3\big(\sigma^2 + 14T\nu^2\big)$$

$$+ 12\alpha_r^2\eta_r^2 L^2 T^2 \bar{E}_r.$$

Finally, apply the factor 2 from equation 21 and add $2\alpha_r^2 \bar{E}_r$. This is precisely the assertion of the lemma. $\square$

**Lemma 4** (Residual recursion under a $\delta$–contractive compressor). *Fix a round $r$ and define $s_r := \eta_r LT \leq \frac{1}{8}$. Let Assumptions 1–3 hold and let the compressor satisfy Definition 1 with parameter $\delta \geq 1$. Define*

$$\bar{E}_r := \frac{1}{K}\sum_{k=1}^{K} \|e_r^{(k)}\|^2, \qquad g_r^{(k)} := \eta_r \sum_{t=0}^{T-1} \big(\nabla f_k(w_{r+\frac{1}{2},t}^{(k)}) + \xi_{k,t}\big).$$

*Then, with all expectations conditional on the randomness inside round $r$, the averaged residual energy obeys*

$$\bar{E}_{r+1} \;\leq\; \rho_r \bar{E}_r \;+\; \Big(1 - \tfrac{1}{\delta}\Big)\Big[ B_r^{(\nabla)} + B_r^{(\nu,\sigma)} \Big], \tag{22}$$

*where*

$$\rho_r = \Big(1 - \tfrac{1}{\delta}\Big)\Big(2(1 - \alpha_r)^2 + 24\,\alpha_r^2 s_r^2\Big) = \Big(1 - \tfrac{1}{\delta}\Big)\Big(2 - 4\alpha_r + (2 + 24s_r^2)\alpha_r^2\Big),$$

*and*

$$B_r^{(\nabla)} := \Big(8\,\eta_r^2 T^2 + 288\,L^2\eta_r^4 T^4\Big)\beta^2\,\|\nabla f(w_r)\|^2,$$

$$B_r^{(\nu,\sigma)} := 8\,\eta_r^2 T^2\,\nu^2 + 4\,\eta_r^2 T\,\sigma^2 + 96\,L^2\eta_r^4 T^3\,\sigma^2 + 1344\,L^2\eta_r^4 T^4\nu^2.$$

*Proof.* Write $u_{r+1}^{(k)} = (1-\alpha_r)e_r^{(k)} + g_r^{(k)}$ and $e_{r+1}^{(k)} = u_{r+1}^{(k)} - \mathcal{C}(u_{r+1}^{(k)})$. By Definition 1,

$$\mathbb{E}_r\|e_{r+1}^{(k)}\|^2 \ \leq\ \left(1-\tfrac{1}{\delta}\right)\mathbb{E}_r\|u_{r+1}^{(k)}\|^2.$$

Averaging over $k$ and using $\|a+b\|^2 \leq 2\|a\|^2 + 2\|b\|^2$ gives

$$\frac{1}{K}\sum_k \mathbb{E}_r\|u_{r+1}^{(k)}\|^2 \ \leq\ 2(1-\alpha_r)^2\,\bar{E}_r \ +\ \frac{2}{K}\sum_k \mathbb{E}_r\|g_r^{(k)}\|^2.$$

Next, using the noise assumptions,

$$\mathbb{E}_r\|g_r^{(k)}\|^2 \leq 2\eta_r^2\,\mathbb{E}_r\Big\|\sum_{t=0}^{T-1}\nabla f_k(w_{r+\frac{1}{2},t}^{(k)})\Big\|^2 + 2\eta_r^2 T\sigma^2.$$

By $L$-smoothness, $\nabla f_k(w_{r+\frac{1}{2},t}^{(k)}) = \nabla f_k(w_r) + \Delta_{k,t}$ with $\|\Delta_{k,t}\| \leq L\|w_{r+\frac{1}{2},t}^{(k)} - w_r\|$, hence

$$\mathbb{E}_r\Big\|\sum_{t=0}^{T-1}\nabla f_k(w_{r+\frac{1}{2},t}^{(k)})\Big\|^2 \leq 2T^2\|\nabla f_k(w_r)\|^2 + 2L^2 T\sum_{t=0}^{T-1}\mathbb{E}_r\|w_{r+\frac{1}{2},t}^{(k)} - w_r\|^2.$$

Averaging over $k$ and using Assumption 3 yields

$$\frac{1}{K}\sum_k \mathbb{E}_r\|g_r^{(k)}\|^2 \leq 4\eta_r^2 T^2\big(\beta^2\|\nabla f(w_r)\|^2 + \nu^2\big) + 4\eta_r^2 L^2 T\, S + 2\eta_r^2 T\sigma^2,$$

where $S := \frac{1}{K}\sum_k \sum_{t=0}^{T-1}\mathbb{E}_r\|w_{r+\frac{1}{2},t}^{(k)} - w_r\|^2$. By Lemma 2 (summed over $t$),

$$S \leq 12\,\eta_r^2 T^2\sigma^2 + 168\,\eta_r^2 T^3\nu^2 + 36\,\eta_r^2 T^3\beta^2\|\nabla f(w_r)\|^2 + 3T\alpha_r^2\bar{E}_r.$$

Substituting and noting $s_r^2 = L^2\eta_r^2 T^2$ gives

$$\frac{1}{K}\sum_k \mathbb{E}_r\|g_r^{(k)}\|^2 \leq \tfrac{1}{2}\big(B_r^{(\nabla)} + B_r^{(\nu,\sigma)}\big) + 12\,\alpha_r^2 s_r^2\,\bar{E}_r.$$

Plugging this into the bound on $\frac{1}{K}\sum_k \mathbb{E}_r\|u_{r+1}^{(k)}\|^2$ yields

$$\frac{1}{K}\sum_k \mathbb{E}_r\|u_{r+1}^{(k)}\|^2 \leq \big(2(1-\alpha_r)^2 + 24\,\alpha_r^2 s_r^2\big)\bar{E}_r + \big(B_r^{(\nabla)} + B_r^{(\nu,\sigma)}\big).$$

Finally, multiplying by $(1-\tfrac{1}{\delta})$ gives equation 22. □

**Corollary 4** (Uniform contraction for $\alpha_r \equiv 1$). *Assume* $\sup_r s_r \leq 1/8$ *and choose* $\alpha_r \equiv 1$. *Then*

$$\rho_{\max} = \sup_r \rho_r \leq \left(1-\tfrac{1}{\delta}\right)24\sup_r s_r^2 \leq 24\cdot(1/8)^2 = \tfrac{3}{8}, \qquad hence \qquad \frac{1}{1-\rho_{\max}} \leq \frac{8}{5}.$$

*In contrast, for the EF baseline* $\alpha_r \equiv 0$, $\rho_{EF} = 2(1-\tfrac{1}{\delta}) \geq 1$ *whenever* $\delta \geq 2$.

**Proposition 1** (Residual contraction vs. EF). *Under the conditions of Lemma 4, let* $\rho_{EF} := 2\big(1-\tfrac{1}{\delta}\big)$ *denote the* $\alpha_r = 0$ *baseline. Then*

$$\rho_r = \left(1-\tfrac{1}{\delta}\right)\left(2 - 4\alpha_r + (2 + 24s_r^2)\alpha_r^2\right), \qquad s_r = \eta_r LT \leq \tfrac{1}{8},$$

*and:*

1. Strict improvement region. *For any $\alpha_r \in \left(0, \frac{2}{1+12s_r^2}\right)$, we have $\rho_r < \rho_{\mathrm{EF}}$, and equality holds at $\alpha_r = \frac{2}{1+12s_r^2}$. Equivalently, the sign of $\rho_r - \rho_{\mathrm{EF}}$ is independent of $\delta$ because $\left(1 - \frac{1}{\delta}\right)$ is a positive multiplicative factor.*

2. Optimal step-ahead. *The minimizer over $\alpha_r \in [0,1]$ is $\alpha_r^\star = \frac{1}{1+12s_r^2} \in (0.84, 1]$, and the minimum value is*
$$\rho_{\min} = \left(1 - \tfrac{1}{\delta}\right)\left(2 - \tfrac{2}{1+12s_r^2}\right) = \rho_{\mathrm{EF}}\left(1 - \tfrac{1}{1+12s_r^2}\right).$$
*In particular, as $s_r \to 0$, $\rho_{\min} \to 0$ and $\rho_{\min}/\rho_{\mathrm{EF}} \to 0$.*

*Proof.* Direct algebra from the quadratic form of $\rho_r$. For (i),
$$\rho_r - \rho_{\mathrm{EF}} = \left(1 - \tfrac{1}{\delta}\right)\left[-4\alpha_r + (2 + 24s_r^2)\alpha_r^2\right] = \left(1 - \tfrac{1}{\delta}\right)\alpha_r\left((2 + 24s_r^2)\alpha_r - 4\right),$$

which is negative for $\alpha_r \in \left(0, \frac{4}{2+24s_r^2}\right) = \left(0, \frac{2}{1+12s_r^2}\right)$ and zero at equality. For (ii), minimizing $q(\alpha) = 2 - 4\alpha + (2 + 24s_r^2)\alpha^2$ gives $\alpha_r^\star = \frac{2}{2+24s_r^2} = \frac{1}{1+12s_r^2}$ and $q(\alpha_r^\star) = 2 - \frac{2}{1+12s_r^2}$; multiplying by $\left(1 - \frac{1}{\delta}\right)$ yields $\rho_{\min}$. $\qquad\square$

## A.3 Partial participation: analysis and rates

At round $r$, let $\mathcal{M}_r \subseteq [K]$ be sampled uniformly without replacement, $|\mathcal{M}_r| = m$, and denote $p := m/K \in (0, 1]$. Write $I_r^{(k)} = \mathbf{1}\{k \in \mathcal{M}_r\}$. Active clients run the same inner loop as in full participation,

$$g_r^{(k)} = \begin{cases} \eta_r \sum_{t=0}^{T-1}\left(\nabla f_k(w_{r+\frac{1}{2},t}^{(k)}) + \xi_{k,t}\right), & I_r^{(k)} = 1, \\ 0, & I_r^{(k)} = 0, \end{cases} \qquad e_{r+1}^{(k)} = \begin{cases} u_{r+1}^{(k)} - C(u_{r+1}^{(k)}), & I_r^{(k)} = 1, \\ e_r^{(k)}, & I_r^{(k)} = 0, \end{cases}$$

with $u_{r+1}^{(k)} = (1 - \alpha_r)e_r^{(k)} + g_r^{(k)}$ for active clients. The server update is

$$w_{r+1} = w_r - \eta\,\bar{C}_{r+1}, \qquad \bar{C}_{r+1} := \frac{1}{m}\sum_{k \in \mathcal{M}_r} C\left(u_{r+1}^{(k)}\right).$$

Expectations $\mathbb{E}_r[\cdot]$ are over local randomness and the draw of $\mathcal{M}_r$. Assumptions 1–3 and Definition 1 (with $\delta \geq 1$) hold.

**Active/global averages; virtual iterate.** Define

$$\tilde{e}_r := \frac{1}{K}\sum_{k=1}^{K} e_r^{(k)}, \qquad \bar{e}_r := \frac{1}{m}\sum_{k \in \mathcal{M}_r} e_r^{(k)}, \qquad \bar{g}_r := \frac{1}{m}\sum_{k \in \mathcal{M}_r} g_r^{(k)}.$$

Note $\mathbb{E}_{\mathcal{M}_r}[\bar{e}_r \mid \{e_r^{(k)}\}] = \tilde{e}_r$. Let $x_r := w_r - \eta\,\tilde{e}_r$.

**Lemma 5** (PP virtual-iterate identity). *With the definitions above,*

$$x_{r+1} - x_r = \eta\left[p\left(\alpha_r\bar{e}_r - \bar{g}_r\right) - (1-p)\,\bar{C}_{r+1}\right]. \tag{23}$$

**Lemma 6** (PP compression second moment). *Let $c_\delta := 2\left(2 - \frac{1}{\delta}\right)$. Then, conditionally on $\mathcal{M}_r$,*

$$\mathbb{E}_r\|\bar{C}_{r+1}\|^2 \leq \frac{1}{m}\sum_{k \in \mathcal{M}_r} \mathbb{E}_r\|C(u_{r+1}^{(k)})\|^2 \leq \frac{c_\delta}{m}\sum_{k \in \mathcal{M}_r} \mathbb{E}_r\|u_{r+1}^{(k)}\|^2.$$

**Lemma 7** (One-round descent under PP). *Let $\Delta_r^{\mathrm{act}} := \alpha_r\bar{e}_r - \bar{g}_r$. Under the same alignment and stepsize coupling as in full participation,*

$$\text{(C1)} \quad \eta_r^2 L^2 \beta^2 (6T^2 + 12T^3) \leq \tfrac{T}{8}, \qquad \text{(C2)} \quad \eta \leq \frac{1}{256\,\beta^2\,L\,\eta_r T},$$

there exists a universal $c > 0$ such that

$$\mathbb{E}_r\big[f(x_{r+1}) - f(x_r)\big] \;\leq\; -c\,p\,\eta\,\eta_r T\,\|\nabla f(w_r)\|^2 \;+\; \mathbb{E}_r\big[\widehat{\mathcal{E}}_r\,\tilde{E}_r\big] \;+\; \widehat{\mathcal{V}}_r\,\sigma^2 \;+\; \widehat{\mathcal{H}}_r\,\nu^2, \tag{24}$$

where $\tilde{E}_r := \frac{1}{K}\sum_{k=1}^{K}\|e_r^{(k)}\|^2$ and $\widehat{\mathcal{E}}_r, \widehat{\mathcal{V}}_r, \widehat{\mathcal{H}}_r$ equal the full-participation coefficients scaled by $p$ and augmented by $(1-p)$–terms originating from $\bar{C}_{r+1}$ via Lemma 6.

**Lemma 8** (Residual recursion under PP). *Let*

$$\rho_r^{\mathrm{PP}} := (1-p) \;+\; p\Big(1 - \tfrac{1}{\delta}\Big)\Big(2(1 - \alpha_r)^2 + 24\,\alpha_r^2(\eta_r LT)^2\Big).$$

*Then*

$$\mathbb{E}_r \tilde{E}_{r+1} \;\leq\; \rho_r^{\mathrm{PP}}\,\tilde{E}_r \;+\; p\Big(1 - \tfrac{1}{\delta}\Big)\Big[B_r^{(\nabla)} + B_r^{(\nu,\sigma)}\Big], \tag{25}$$

*with $B_r^{(\nabla)}$ and $B_r^{(\nu,\sigma)}$ as in full participation.*

**Telescoping, absorption, and final bounds (PP).** Let $S_R := \sum_{r=0}^{R-1}\eta_r T$ and $S_R^{\mathrm{PP}} := \sum_{r=0}^{R-1} p\,\eta_r T = p\,S_R$. Summing equation 24 over $r$ and using $\mathcal{C}_r^{\mathrm{PP}} \leq -c\,p\,\eta\,\eta_r T$,

$$\sum_{r=0}^{R-1} p\,\eta_r T\,\mathbb{E}\|\nabla f(w_r)\|^2 \;\leq\; \frac{16}{\eta}\big(f(x_0) - \mathbb{E}f(x_R)\big) + \frac{16}{\eta}\sum_{r=0}^{R-1}\mathbb{E}\big[\widehat{\mathcal{E}}_r\,\tilde{E}_r + \widehat{\mathcal{V}}_r\,\sigma^2 + \widehat{\mathcal{H}}_r\,\nu^2\big].$$

If $f$ is bounded below by $f_\star$, then $f(x_R) \geq f_\star$. Summing equation 25 and assuming $\rho_{\max}^{\mathrm{PP}} := \sup_r \rho_r^{\mathrm{PP}} < 1$,

$$\sum_{r=0}^{R-1}\mathbb{E}\tilde{E}_r \;\leq\; \frac{1}{1 - \rho_{\max}^{\mathrm{PP}}}\,\tilde{E}_0 + \frac{p}{1 - \rho_{\max}^{\mathrm{PP}}}\Big(1 - \tfrac{1}{\delta}\Big)\sum_{r=0}^{R-1}\mathbb{E}\big[B_r^{(\nabla)} + B_r^{(\nu,\sigma)}\big].$$

Plugging this into the previous display yields a gradient-forcing term proportional to

$$\frac{16}{\eta} \times \frac{\widehat{\mathcal{E}}_{\max}}{1 - \rho_{\max}^{\mathrm{PP}}}\Big(1 - \tfrac{1}{\delta}\Big) \times \underbrace{p\sum_r B_r^{(\nabla)}}_{\leq\,\beta^2 d_{\max}\,p\sum_r \eta_r T\,\mathbb{E}\|\nabla f(w_r)\|^2},$$

where $d_{\max} := \sup_r\big(8\,\eta_r T + 288\,L^2\eta_r^3 T^3\big)$. Defining

$$\Theta_{\mathrm{PP}} := \frac{16}{\eta} \times \frac{\widehat{\mathcal{E}}_{\max}}{1 - \rho_{\max}^{\mathrm{PP}}}\Big(1 - \tfrac{1}{\delta}\Big)\beta^2 d_{\max},$$

we obtain the absorption inequality (the factor $p$ cancels on both sides):

$$\big(1 - \Theta_{\mathrm{PP}}\big)\sum_{r=0}^{R-1} p\,\eta_r T\,\mathbb{E}\|\nabla f(w_r)\|^2 \;\leq\; \frac{16}{\eta}\Big[(f(x_0) - f_\star) + \sum_{r=0}^{R-1}\widehat{\mathcal{V}}_r\,\sigma^2 + \widehat{\mathcal{H}}_r\,\nu^2\Big].$$

Assuming $\Theta_{\mathrm{PP}} \leq \frac{1}{2}$, we conclude

$$\sum_{r=0}^{R-1} p\,\eta_r T\,\mathbb{E}\|\nabla f(w_r)\|^2 \;\leq\; \frac{32}{\eta}\Big[(f(x_0) - f_\star) + \sum_{r=0}^{R-1}\widehat{\mathcal{V}}_r\,\sigma^2 + \widehat{\mathcal{H}}_r\,\nu^2\Big].$$

Dividing by $S_R^{\mathrm{PP}} = p\,\eta_0 TR$ and with $\eta_r \equiv \eta_0$ yields the averaged bounds below.

$$\frac{1}{R}\sum_{r=0}^{R-1}\mathbb{E}\|\nabla f(w_r)\|^2 \;\leq\; \frac{32}{\eta\,p\,\eta_0 TR}\,(f(x_0) - f_\star) \;+\; \frac{128\,L\,\eta_0}{\eta\,p\,m}\,\sigma^2$$
$$+\; \frac{32}{\eta\,p}\Big(1 - \tfrac{1}{\delta}\Big)\Big[C_\sigma\,\eta_0^2 L^2 T\,\sigma^2 + C_\nu\,\eta_0^2 L^2 T^2\,\nu^2\Big]. \tag{26}$$

So, the optimization term scales as $O\big((p\,\eta\,\eta_0 TR)^{-1}\big)$ (a per-round slow-down by $1/p = K/m$). The *pure mini-batch* variance enjoys a $1/m$ reduction: its contribution scales as $\frac{128\,L\,\eta_0}{m}\,\sigma^2$, while the residual-induced variance/heterogeneity floors scale as $\big(1 - \tfrac{1}{\delta}\big)[C_\sigma\,\eta_0^2 L^2 T\,\sigma^2 + C_\nu\,\eta_0^2 L^2 T^2\nu^2]$.

**Stalling vs. step-ahead.** With $\alpha_r = 0$ (no step-ahead), the multiplicative factor becomes $\rho_r^{\mathrm{PP}} = (1 - p) + 2p(1 - \frac{1}{\delta}) = 1 + p(1 - \frac{2}{\delta})$, which can be $\geq 1$ under aggressive compression and small $p$, explaining the slowdown in cross-device regimes. For moderate $\alpha_r$ (e.g., $\alpha_r \approx \alpha_r^{\star}$ from the full-participation analysis), $\rho_r^{\mathrm{PP}}$ strictly decreases, improving the decay of $\tilde{E}_r$ each time a client participates and restoring faster early progress.

**Constants and feasibility.** The constants $C_\sigma$, $C_\nu$ and $\Theta$ in Theorem 1 collect the contributions of stochastic-gradient variance, data heterogeneity, and compression–induced residual drift. Inspecting their explicit formulas, we see that they depend on the algorithmic hyperparameters only through the effective local stepsize

$$s_0 = \eta_0 LT,$$

the compression bias factor $(1 - 1/\delta)$, and the residual–contraction term $1/(1 - \rho_{\max})$. In particular, $\rho_{\max}$ itself is an increasing function of $s_0$ and $(1 - 1/\delta)$, so $C_\sigma$, $C_\nu$ and $\Theta$ are *monotone nondecreasing* in $s_0$ and $(1 - 1/\delta)$: larger local work or more aggressive compression lead to larger constants and thus a higher residual-driven floor.

In the partial-participation extension, the corresponding constant $\Theta_{\mathrm{PP}}$ inherits the same monotone dependence on $s_0$ and $(1 - 1/\delta)$ and, in addition, scales inversely with the participation rate $p = m/K$ (i.e., it increases as $p$ decreases). Thus, the feasibility conditions $\rho_{\max}^{\mathrm{PP}} < 1$ and $\Theta_{\mathrm{PP}} \leq \frac{1}{2}$ can be interpreted as requiring a standard "small" effective local stepsize $s_0$, moderate compression, and not-too-extreme partial participation. For representative hyperparameters used in our experiments (e.g., $T = 5$, Top-$k$ compression, and $p \in \{0.1, 0.5, 1.0\}$), we numerically evaluate these constants and confirm that $\rho_{\max}^{\mathrm{PP}} < 1$ in all regimes. Moreover, in a mildly compressed setting (e.g., $\delta = 1.005$ with the same stepsizes), the corresponding $\Theta_{\mathrm{PP}}$ lies well below $\frac{1}{2}$, illustrating that the condition $\Theta_{\mathrm{PP}} \leq \frac{1}{2}$ is a conservative sufficient condition rather than a tight practical tuning rule for the aggressively compressed Top-$k$ regimes we study.

### A.4 Comparison of EF-type compressed FL methods

For completeness, In Table 2, we summarize the main assumptions, mechanisms, and qualitative nonconvex behavior of several closely related algorithms: Fed-EF, SAEF, CSER, SCAFCOM, and SA-PEF. The goal is not to restate full theorems, but to highlight the regimes they target. Fed-EF and SA-PEF share the same lightweight, stateless EF architecture and standard FL assumptions (local steps, partial participation, biased contractive compressors), with SA-PEF improving the residual-contraction constant under compression. CSER and SAEF focus on centralized/local-SGD settings without partial participation, while SCAFCOM achieves stronger robustness to arbitrary heterogeneity by adding SCAFFOLD-style control variates and momentum, at the cost of increasing per-client state and communication.

**Remark 2** (Relation to EF21). *EF21 and its extensions (Richtárik et al., 2021; Fatkhullin et al., 2025) obtain stronger guarantees (no error floor) under a different regime: synchronized data-parallel training with T=1, full-gradient (or gradient-difference) compression at a shared iterate, and no local steps. In this setting, EF21 is strictly preferable to classical EF. Our analysis targets a complementary regime, federated local-SGD with T > 1 local steps, partial participation, and biased contractive compressors, where the compressed object is the accumulated local update and client drift plays a central role. Extending EF21-style arguments (or designing EF21-style step-ahead variants) to this local-SGD, partial-participation, biased-compression setting is non-trivial and remains an interesting direction for future work.*

## B Implementation Details and Additional Experiments

### B.1 Setup and Parameter Tuning

This appendix details datasets, federated partitioning, compressors, the hyperparameter search protocol, and fairness controls used across all methods.

**Datasets, models, and preprocessing.** CIFAR-10 (ResNet-9), CIFAR-100 (ResNet-18), and Tiny-ImageNet (64×64; ResNet-34) are trained with cross-entropy loss. Preprocessing follows standard practice:

Table 2: Comparison of compressed algorithms for FL. SA-PEF bridges the gap between Fed-EF (stable but slower under aggressive compression) and SAEF (faster warm-up but fragile in heterogeneous FL), achieving improved residual contraction without the extra state and complexity of control-variate methods such as SCAFCOM.

| Algorithm | Assumptions (beyond $L$-smoothness) | Mechanism & State | Convergence / Behavior (Nonconvex) |
|---|---|---|---|
| Fed-EF (Li & Li, 2023) | Bounded variance; local SGD with PP | Biased $\delta$-contractive; Residual-only state | Standard FL nonconvex rate with $1/p$ slowdown under partial participation; residual contraction factor $\rho_{EF}$ can be relatively weak under aggressive compression, leading to slower progress and earlier stalling. |
| SAEF (Xu et al., 2021) | Bounded gradient; centralized, synchronous setting; no PP analysis | Full step-ahead ($\alpha = 1$); Residual-only state | Analyzed in the classical distributed setting (no local steps, no client sampling); reduces EF's gradient mismatch there. In our FL experiments with heterogeneous data, full step-ahead tends to produce larger gradient mismatch and late-stage plateaus compared to EF/SA-PEF (Sec. 4). |
| CSER (Xie et al., 2020) | Bounded variance; local SGD (typically full participation) | Error reset (periodic dense communication of residuals); Residual-only state | Controls residual drift via periodic resets, yielding a nonconvex local-SGD rate with an $R$-independent floor. However, resets require sending full residuals, inducing **high peak bandwidth** at reset rounds and analyses usually assume full participation. |
| SCAFCOM (Huang et al., 2024) | **Arbitrary heterogeneity**; bounded variance; local SGD with PP | Control variates + momentum; **Stateful** (extra per-client state $c$, $c_i$) | Achieves nonconvex FL guarantees under arbitrary non-IID data, with improved dependence on heterogeneity, at the cost of **higher system complexity** (maintaining control-variates state). |
| **SA-PEF (ours)** | Gradient dissimilarity $(\beta, \nu)$; local SGD with PP | Partial step-ahead $(0 < \alpha < 1)$; Residual-only state | Operates in the same FL regime and under the same assumptions as Fed-EF, with the same leading-order nonconvex rate, but with a **strictly improved** residual contraction factor $\rho_{\max} < \rho_{EF}$ under biased compression, which lowers the error floor and balances warm-up speed with long-term stability. |

per-dataset mean/std normalization; CIFAR uses random crop (4-pixel padding) and horizontal flip; Tiny-ImageNet uses random resized crop to 64 and horizontal flip. Unless stated otherwise, the batch size is 64, momentum is 0.9, weight decay is $5 \times 10^{-4}$ on CIFAR and $10^{-4}$ on Tiny-ImageNet.

**Federated partitioning, participation, and local computation.** We create $K{=}100$ clients and apply Dirichlet label partitioning with $\gamma \in \{0.1, 0.5, 1.0\}$ (smaller $\gamma \Rightarrow$ stronger non-IID). Each round samples $m = \lfloor pK \rfloor$ clients uniformly without replacement with $p \in \{0.1, 0.5, 1.0\}$. Participating clients run $T$ local SGD steps at stepsize $\eta_r$; default $T{=}5$. We train for $R{=}200$ rounds. Unless stated, server stepsize is $\eta{=}1.0$. We conducted all federated learning simulations using the FLOWER framework (Beutel et al., 2020).

**Compressors and communication accounting.** We use Top-$k$ sparsification with $k/d \in \{0.01, 0.05, 0.10\}$; each selected entry communicates its *index* and *value*. As a consequence, Top-$k$ satisfies Definition 1 with $\delta = d/k$; we record the standard bound below.

**Lemma 9** (Top-$k$ contraction; (Stich et al., 2018; Beznosikov et al., 2023)). *Let $C = \mathrm{Top}_k : \mathbb{R}^d \to \mathbb{R}^d$ keep the $k$ largest absolute-value coordinates of $x$ (ties broken arbitrarily), zeroing the rest. Then for all $x \in \mathbb{R}^d$,*

$$\|x - C(x)\|_2^2 \;\le\; \left(1 - \frac{k}{d}\right) \|x\|_2^2, \qquad \frac{k}{d} \|x\|_2^2 \;\le\; \|C(x)\|_2^2 \;=\; \langle C(x), x \rangle \;\le\; \|x\|_2^2.$$

*In particular, $C$ is $\delta$-contractive with $\delta = d/k$ in the sense of Definition 1. The constants are tight when all $|x_i|$ are equal.*

Each selected entry transmits its *index* and *value*. *Raw uplink bits* per participating client per round are $k\big(\lceil \log_2 d \rceil + b_{\mathrm{val}}\big)$ with $b_{\mathrm{val}}{=}32$ for FP32 values. Unless stated, reported *cumulative communication* aggregates *uplink only* across participating clients (downlink is identical across compressed methods and omitted for fairness); FedAvg's downlink/uplink are both dense FP32.

**Hyperparameter search protocol.** We adopt a small, method-agnostic grid tuned on a held-out validation split. Unless noted, we select hyperparameters by best *validation top-1* at a *fixed communication budget* (bits) within $R$ rounds; ties are broken by higher accuracy at earlier checkpoints. We reuse the *same* grid across participation rates ($p$).

**Search spaces (shared across methods).**

- **Client LR $\eta_r$:**

  | | |
  |---|---|
  | CIFAR-10: | $\{0.001, 0.05, 0.1, 0.2, 1.0, 10\}$ |
  | CIFAR-100: | $\{0.001, 0.05, 0.1, 1.0, 10\}$ |
  | Tiny-ImageNet: | $\{0.001, 0.02, 0.05, 1.0\}$ |
  | | (cosine decay with 3-5 epoch warm-up, minimum $\eta_r = 0.005$) |

- **Server LR $\eta$:** $\{0.5, 1.0\}$.

- **Weight decay:** $\{5 \times 10^{-4}, 10^{-4}\}$.

- **Local steps $T$:** $\{1, 5, 10\}$.

- **Compressor level $k/d$:** $\{0.01, 0.05, 0.10\}$.

**SA-PEF-specific.** Constant-$\alpha$ ablations use $\alpha \in \{0.0, 0.1, 0.2, 0.3, 0.4, 0.5, 0.6, 0.7, 0.8, 0.9, 1.0\}$; unless stated, $\alpha$ is fixed across rounds.

**Fairness controls and evaluation.** (i) The *same* grid is used across methods; (ii) the best setting is selected at a matched bit budget; (iii) client sampling seeds are shared across methods; (iv) evaluation uses the server model in `eval` mode with identical preprocessing. We report both *accuracy vs. rounds* and *accuracy vs. bits*; the latter is our primary metric under communication constraints. Experiments ran on NVIDIA A100/A5000/H200 GPUs; hardware does not affect communication accounting.

Table 3: Final test accuracy (mean $\pm$ std. over five independent runs) for CIFAR-10 and CIFAR-100 under two participation/heterogeneity regimes. The hyperparameters used in all algorithms are $R = 200$, $T = 5$, and $\eta_l = 0.1$.

| Dataset | Model | Algorithm | Final test accuracy (%) | | | |
|---|---|---|---|---|---|---|
| | | | $p = 0.1,\ \gamma = 0.5$ | | $p = 0.5,\ \gamma = 0.1$ | |
| | | | top-1 | top-10 | top-1 | top-10 |
| CIFAR-10 | ResNet-9 | FedAvg | $88.5 \pm 1.6$ | | $69.5 \pm 1.5$ | |
| | | EF | $57.0 \pm 2.0$ | $72.7 \pm 2.3$ | $40.3 \pm 3.7$ | $47.2 \pm 4.1$ |
| | | SAEF | $74.2 \pm 3.9$ | $75.5 \pm 2.9$ | $39.5 \pm 4.6$ | $48.5 \pm 3.3$ |
| | | CSER | $68.6 \pm 2.8$ | $80.5 \pm 1.2$ | $49.5 \pm 4.0$ | $67.2 \pm 2.6$ |
| | | SA-PEF | $80.5 \pm 2.6$ | $82.7 \pm 1.6$ | $47.5 \pm 3.6$ | $68.5 \pm 1.9$ |
| CIFAR-100 | ResNet-18 | FedAvg | $62.5 \pm 1.6$ | | $61.9 \pm 0.6$ | |
| | | EF | $40.4 \pm 2.4$ | $49.5 \pm 1.6$ | $41.5 \pm 1.6$ | $57.5 \pm 1.9$ |
| | | SAEF | $46.1 \pm 1.4$ | $50.5 \pm 2.0$ | $44.5 \pm 3.6$ | $57.5 \pm 3.9$ |
| | | CSER | $46.2 \pm 0.4$ | $51.5 \pm 1.9$ | $42.5 \pm 2.4$ | $60.7 \pm 1.0$ |
| | | SA-PEF | $49.6 \pm 1.8$ | $54.5 \pm 2.6$ | $48.5 \pm 2.0$ | $60.6 \pm 2.8$ |

## B.2 Additional Experiments

**Multi-seed stability.** In Table 3, we report final test accuracy as mean $\pm$ standard deviation over five independent runs with different random seeds. Across seeds, the relative ranking of methods is consistent, with SA-PEF retaining its advantage in high-compression, low-participation regimes.

Figures 5–8 report extra convergence curves for SA-PEF and the baselines on **CIFAR-10**, **CIFAR-100**, and **Tiny-ImageNet**. For each dataset we sweep (i) *participation $p \in \{1.0, 0.1\}$*, (ii) *compression budget* (Top-1% and Top-10% under full participation; Top-5% and Top-1% under $p$=0.1), and (iii) *data heterogeneity* via Dirichlet partitions. The plots include both *accuracy vs. rounds* and *accuracy vs. communicated GB*. Under full participation, SA-PEF consistently reaches a given accuracy in fewer rounds than EF/CSER and tracks SAEF without late-stage plateaus. Under partial participation with aggressive compression (Top-5%, Top-1%), the gaps naturally narrow but the qualitative trend persists, illustrating that the main conclusions are robust across datasets, architectures, and federation settings.

**IID control experiments.** To isolate the effect of data heterogeneity, we also evaluate under IID partitions in Figure 9. We report *accuracy vs. rounds* and *accuracy vs. GB* for: (i) partial participation ($p$=0.5) with Top-5% and Top-10%. Across datasets, SA-PEF matches or exceeds EF/CSER in early rounds under the same communication budget.

**Extreme partial participation and local work.** To further stress-test our methods, we also consider more demanding FL regimes with very low participation and larger local work. In particular, we run experiments with participation $p = 0.05$ and $T = 10$ local SGD steps per round, comparing EF, CSER, and SA-PEF under the same compression level. As shown in Fig. 10, SA-PEF consistently attains higher accuracy than EF and CSER at a fixed communication budget, indicating that its advantages persist even under extreme partial participation and increased local work.

**Wall-clock efficiency.** To quantify implementation overhead, we report test accuracy versus wall-clock time on CIFAR-10/ResNet-9 under a fixed hardware setup (six NVIDIA RTX A5000 GPUs across 6 nodes) in Figure 11. SA-PEF reaches a given accuracy level substantially earlier than EF, SAEF, CSER, and FedAvg, confirming that the small extra vector operations it introduces incur negligible runtime cost while improving time-to-accuracy.

**Scaled-sign compressor.** Besides Top-$k$, we also consider a scaled-sign compressor $C(x) = \frac{\|x\|_1}{d}\, \mathrm{sign}(x)$. This is the group-scaled sign compressor of Li & Li (2023), specialized to a single block ($M = 1$), and

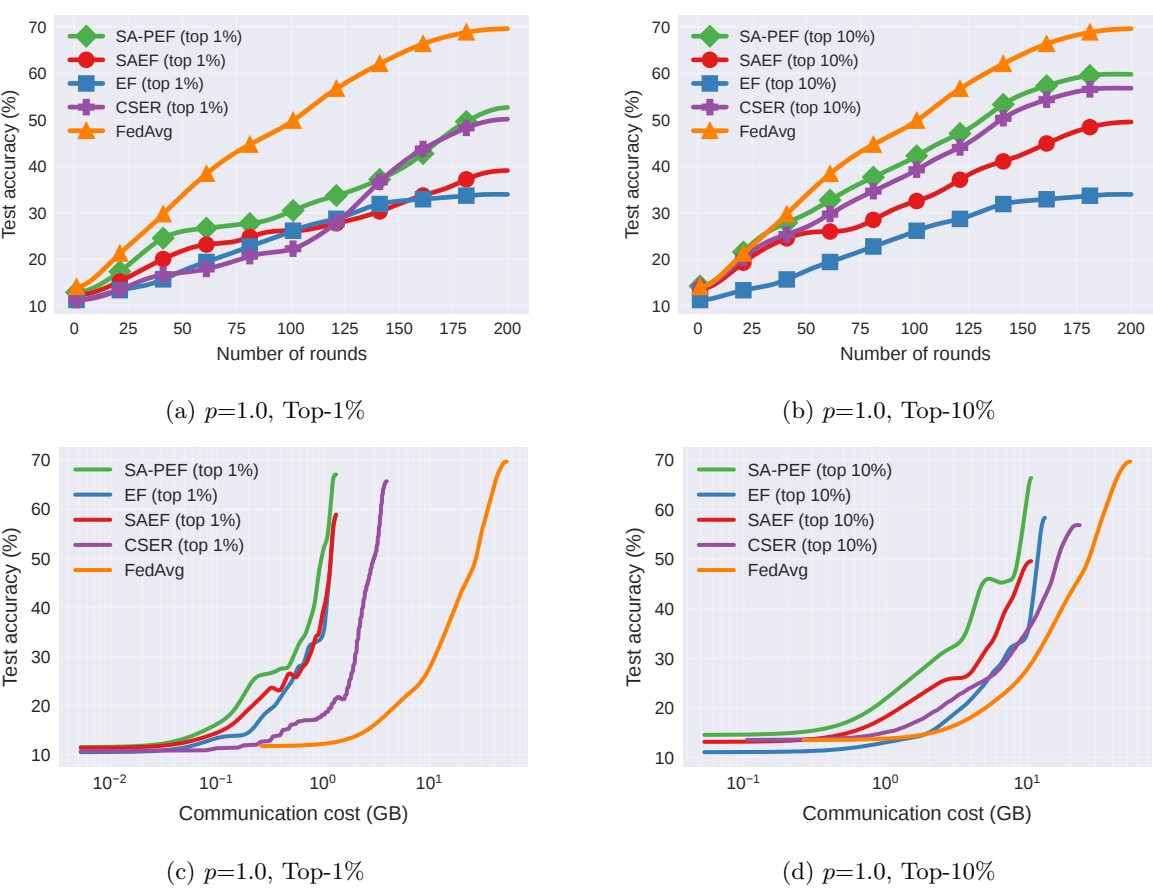

Figure 5: Test accuracy vs. number of rounds (row 1) and communicated GB (row 2) on the CIFAR-10 dataset using ResNet-9 and $\gamma$=0.1.

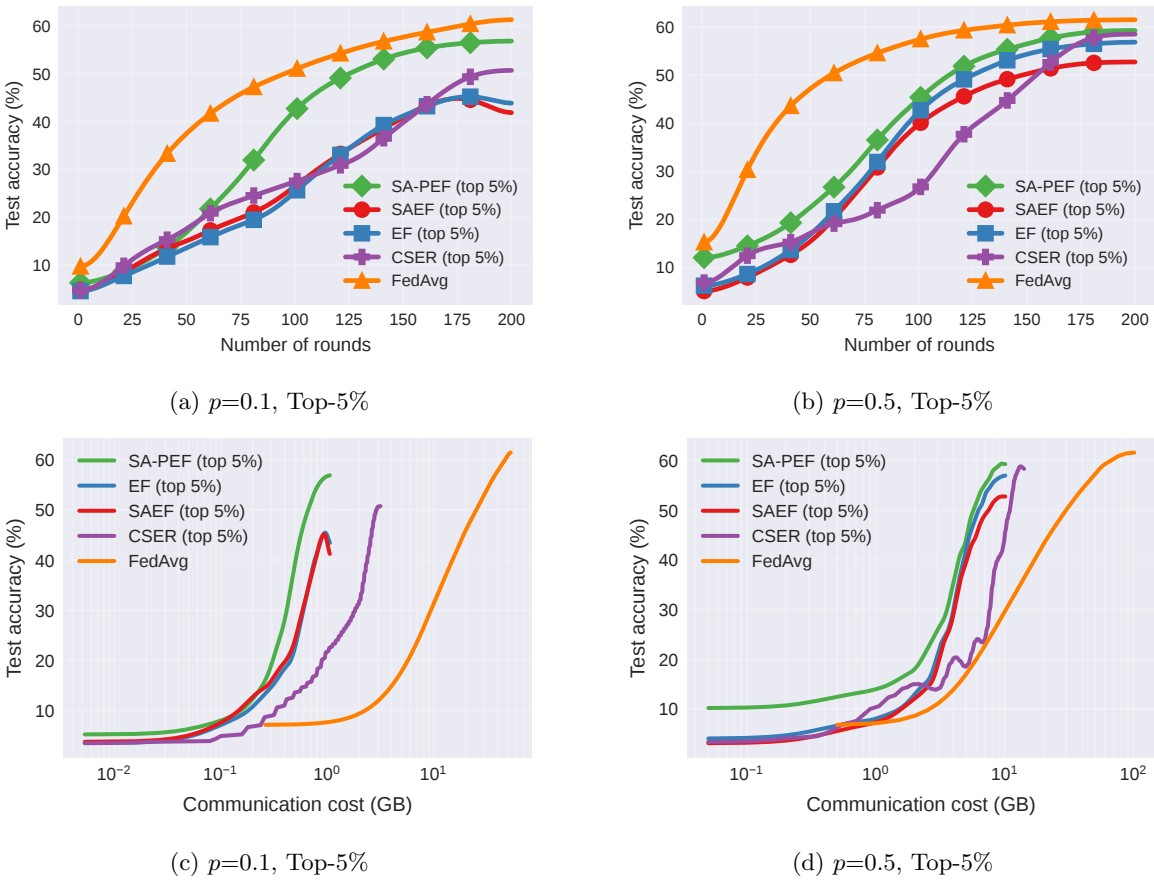

Figure 6: Test accuracy vs. number of rounds (row 1) and communicated GB (row 2) on the CIFAR-100 dataset using ResNet-18 and $\gamma$=0.1.

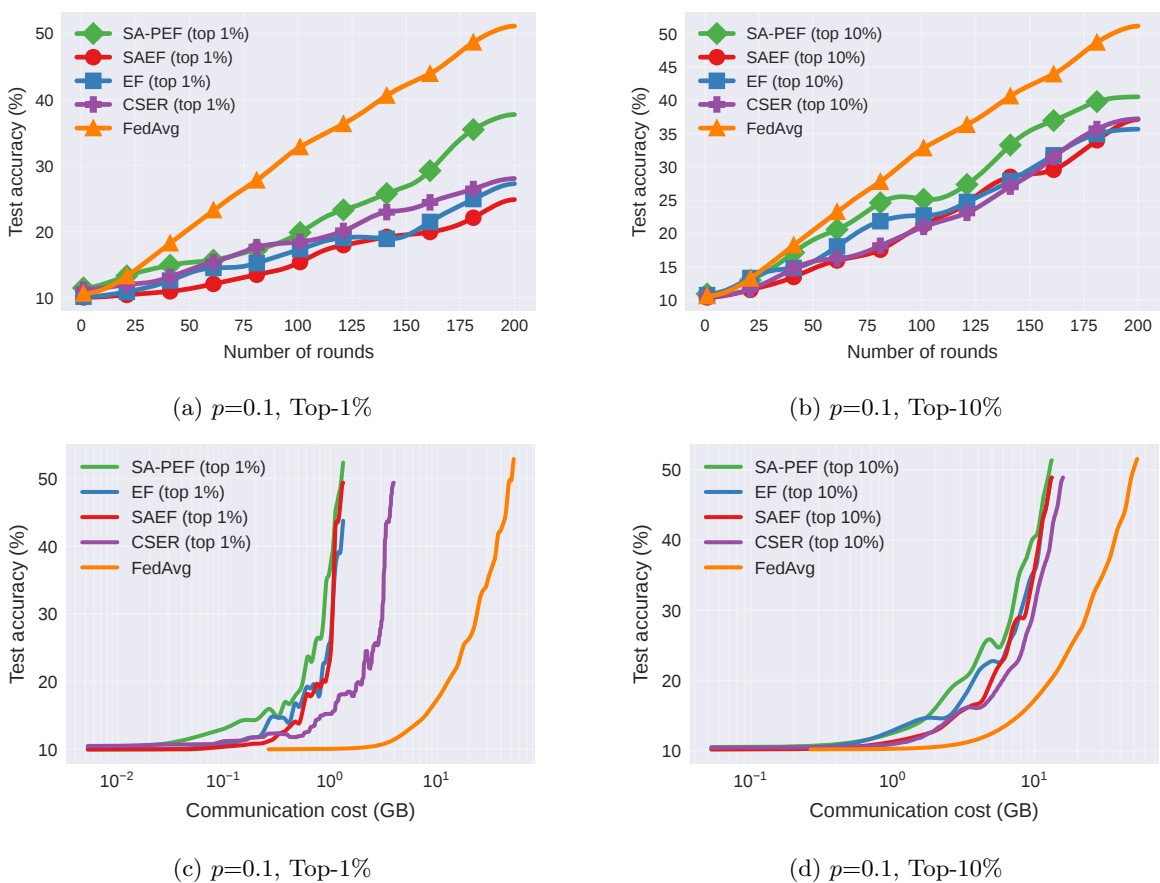

(a) $p$=0.1, Top-1%             (b) $p$=0.1, Top-10%

(c) $p$=0.1, Top-1%             (d) $p$=0.1, Top-10%

Figure 7: Test accuracy vs. number of rounds (row 1) and communicated GB (row 2) on the CIFAR-10 dataset using ResNet-9 and $\gamma$=0.1.

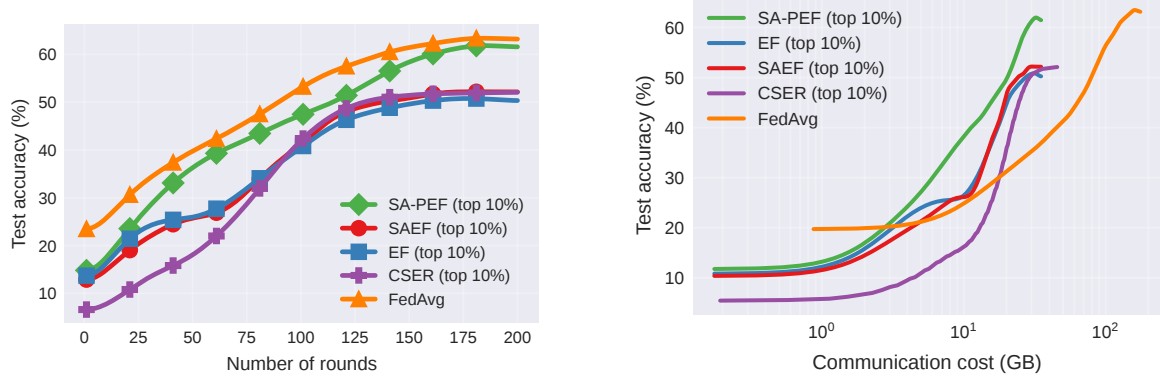

Figure 8: Test accuracy vs. number of rounds (left) and communicated GB (right) on the Tiny-ImageNet dataset using ResNet-34 with $\gamma$=0.5, $p$=0.1.

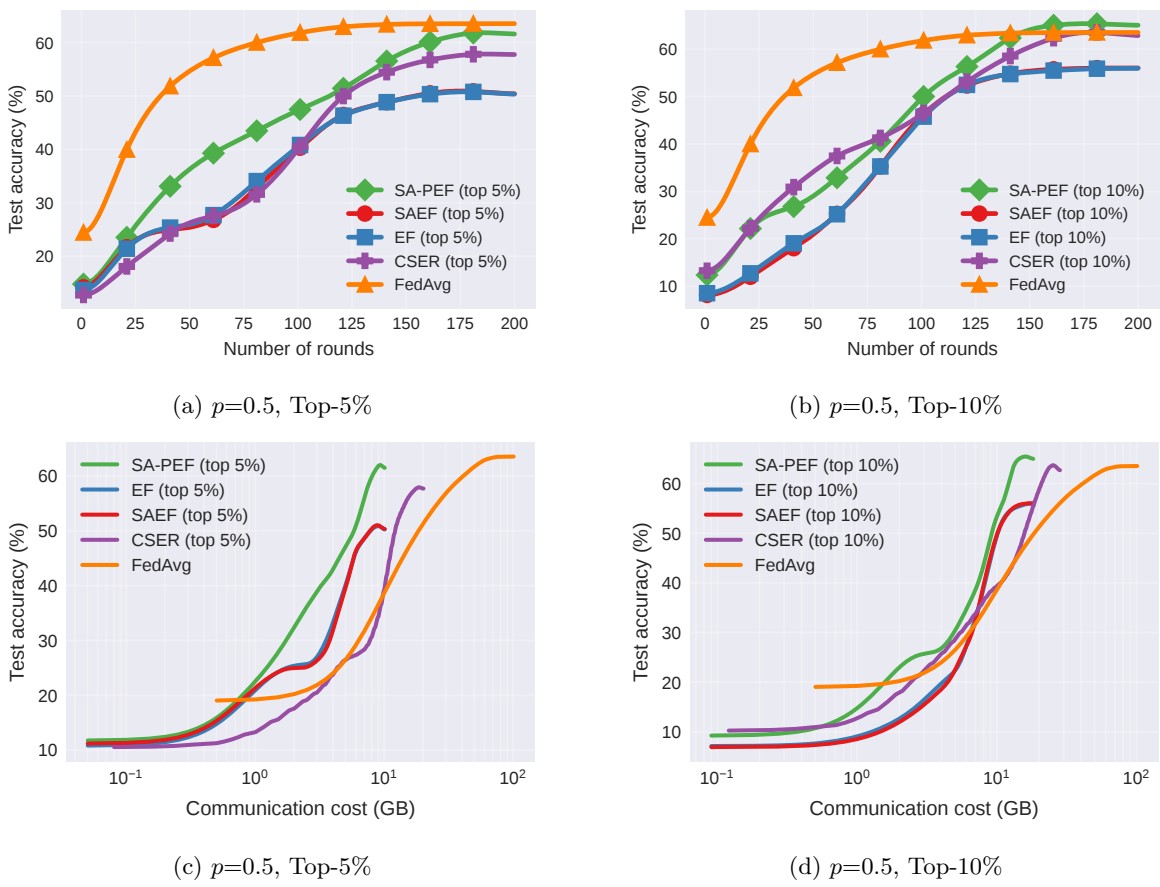

(a) $p$=0.5, Top-5%

(b) $p$=0.5, Top-10%

(c) $p$=0.5, Top-5%

(d) $p$=0.5, Top-10%

Figure 9: Test accuracy vs. number of rounds (row 1) and communicated GB (row 2) on the CIFAR-100 dataset using ResNet-18.

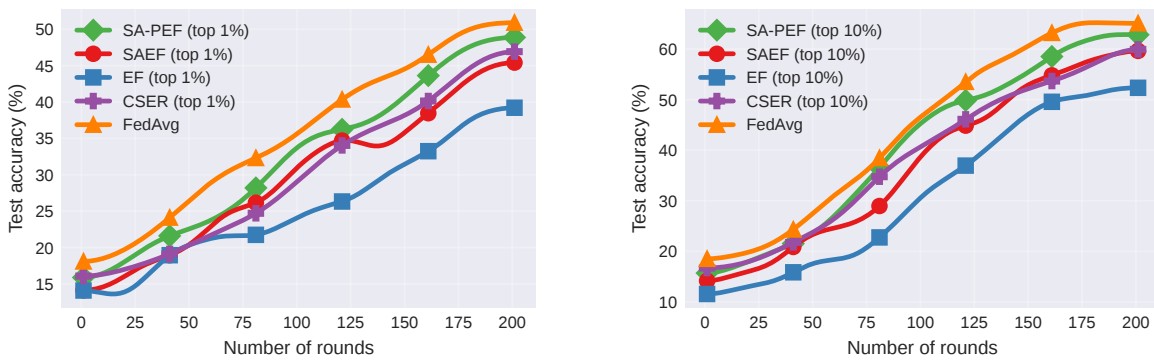

Figure 10: Test accuracy vs. number of rounds on CIFAR-10 with ResNet-9 under Top-1% (left) and Top-10% (right) uplink compression.

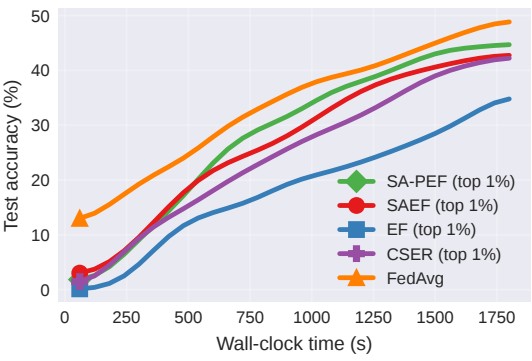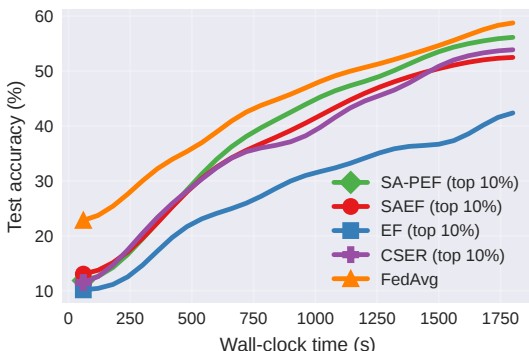

Figure 11: Test accuracy vs. wall-clock time (s) on CIFAR-10 with ResNet-9 under Top-1% (left) and Top-10% (right) uplink compression.

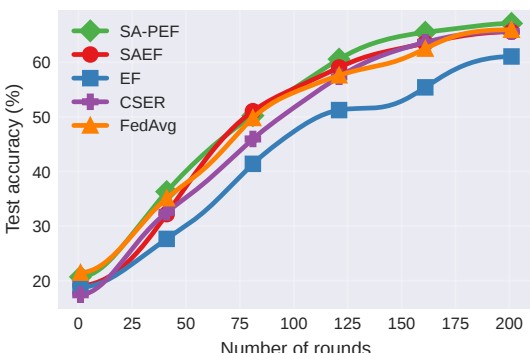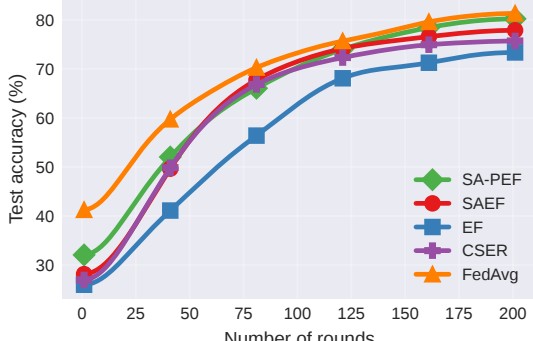

Figure 12: Test accuracy vs. number of rounds on CIFAR-10 with ResNet-9 for Dirichlet-$\gamma$ partitions: $\gamma = 0.1$ (left) and $\gamma = 0.5$ (right).

Proposition C.1 in that work shows that it is contractive in the sense of our Definition 1 for a suitable $\delta > 0$, so it falls within our theoretical framework. Hence, we report preliminary CIFAR-10/ResNet-9 results with this scaled-sign compressor in Figure 12, where SA-PEF again achieves higher accuracy than EF, SAEF, and CSER at a fixed communication budget.

**Effect of momentum.** To isolate the role of momentum, we repeat our CIFAR-10/100 experiments using SGD *without* momentum, keeping all other hyperparameters and compression settings fixed, and provide the results in Figure 13. The test-accuracy trajectories and accuracy-communication curves show that SA-PEF consistently matches or outperforms EF, SAEF, and CSER, and remains close to FedAvg in terms of accuracy per communicated GB. This suggests that our conclusions are essentially momentum-agnostic: momentum slightly reshapes the trajectories but does not drive the gains of SA-PEF over EF-style baselines.

**Comparison with SCAFCOM.** To further assess SA-PEF, we follow the MNIST setup of Huang et al. (2024): a 2-layer fully-connected network is trained on MNIST distributed across $N = 200$ clients in a highly heterogeneous regime (each client holds data from at most two classes), with partial participation $p = 0.1$ and 10 local steps. We apply aggressive Top-1% compression to EF-style methods. As shown in Figure 14, SA-PEF closely tracks SCAFCOM and both substantially outperform standard EF and FedAvg, while uncompressed SCAFFOLD lies in between. This indicates that, under the same communication budget, SA-PEF can match the robustness of SCAFCOM's control-variate-plus-momentum design while retaining the simpler EF architecture (one residual per client, no additional drift-correction state).

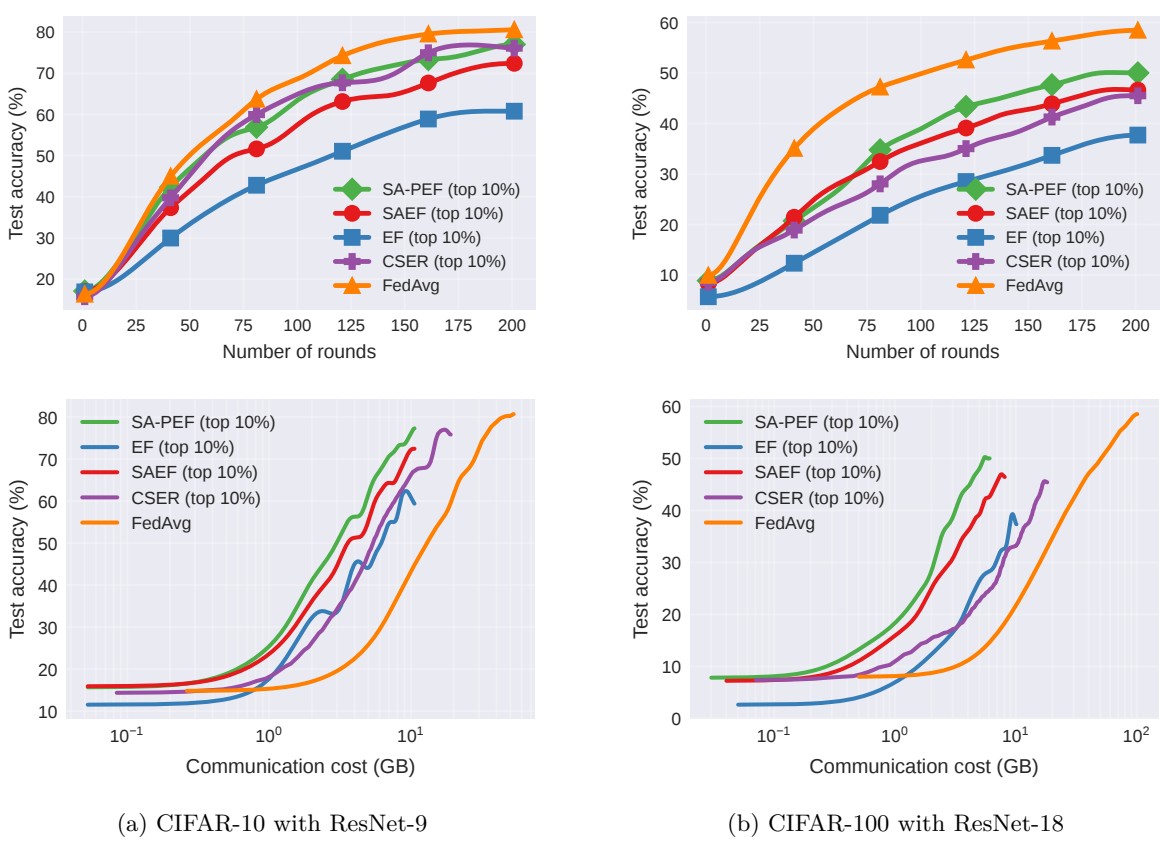

(a) CIFAR-10 with ResNet-9             (b) CIFAR-100 with ResNet-18

Figure 13: Test accuracy vs. number of rounds (row 1) and communicated GB (row 2) using $p$=0.1, Top-10%, and $\gamma$=0.5.

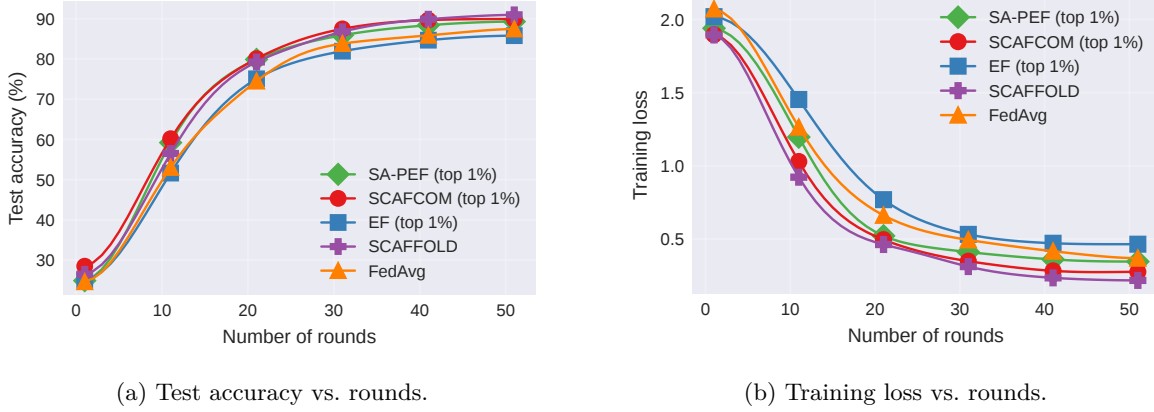

(a) Test accuracy vs. rounds.             (b) Training loss vs. rounds.

Figure 14: Comparison with SCAFCOM on MNIST under Top-1% compression, partial participation $p$=0.1, and 10 local steps.

Table 4: Optimal global ($\eta$) and local ($\eta_r$) learning rate combinations.

| Algorithm | $\eta$ | $\eta_r$ |
|-----------|--------|----------|
| SA-PEF | 1 | $10^{-3}$ |
| SAEF | 1 | $10^{-3}$ |
| EF | 1 | $10^{-3}$ |
| SCAFCOM | 3 | $10^{-1}$ |
| FedAvg | 1 | $10^{-1}$ |

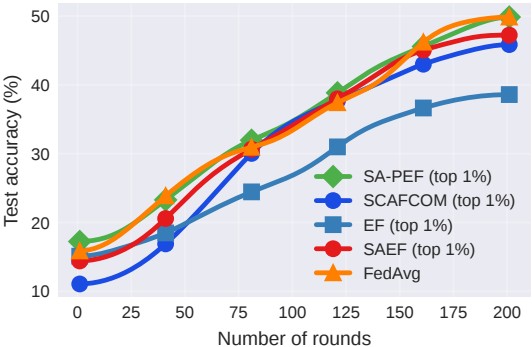 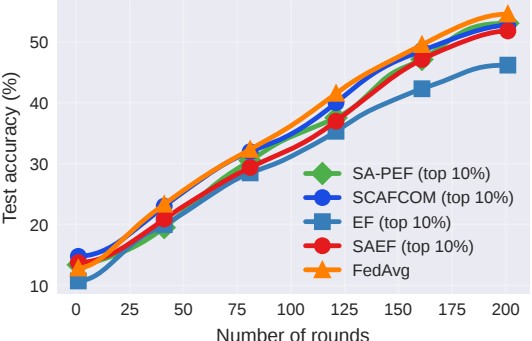

Figure 15: Test accuracy vs. number of rounds on CIFAR-10 with ResNet-9, partial participation $p$=0.1, and 10 local steps under Top-1% (left) and Top-10% (right) uplink compression.

To better align with our main experimental setup, we additionally report a preliminary CIFAR-10/ResNet-9 experiment under aggressive compression in Figure 15. We follow the local mini-batch step formulation of Huang et al. (2024) (rather than local epochs) for a fair comparison. We use $K = 100$ clients and $R = 200$ communication rounds; at each round, the server samples 10 clients, and each selected client performs 20 local mini-batch SGD steps on a ResNet-9 model. For all EF-style methods (EF, SAEF, SA-PEF, and SCAFCOM) we apply Top-1% and Top-10% sparsification to the uplink updates, while FedAvg communicates dense updates. SCAFCOM uses control-variate and momentum coefficients $\alpha_{\text{sc}} = 0.1$ and $\beta_{\text{sc}} = 0.2$, selected via a small grid search. SA-PEF again behaves competitively with SCAFCOM while clearly improving over EF, SAEF, and FedAvg under the same communication budget. All curves are averaged over five independent runs with different random seeds. The learning-rate combinations that yield the highest test accuracy are listed in Table 4.

### B.2.1 Evaluation on FEMNIST

We evaluate SA-PEF on FEMNIST, which provides a writer-partitioned federated benchmark. Our goal is to assess SA-PEF under natural client structure and to study how the relative behavior of EF, SAEF, and SA-PEF changes as heterogeneity becomes more severe.

FEMNIST is partitioned naturally by writer, with roughly 3,500 writers. Since our stateful compressed baselines maintain per-client residual state, simulating all writers simultaneously would incur substantial memory and computational overhead. Following common benchmark practice for large-scale federated evaluations, e.g., pFL-Bench (Chen et al., 2022), we subsample a fixed set of 200 writers per run and sample 20 clients per round. This preserves the writer-partitioned structure while keeping the experiments tractable. We use a common setup across all methods. Following the LEAF benchmark (Caldas et al., 2018), we use a 2-layer CNN and run 200 communication rounds with 200 total clients and 20 sampled per round. Training uses SGD (momentum 0.9, weight decay $10^{-4}$, batch size 32) for 5 local epochs. Compression uses Top-$k$ with $k = 1\%$ and error feedback, except for FedAvg, which is dense. For SA-PEF, we use $\alpha_r \equiv 0.85$. We report the results as mean test accuracy over three seeds.

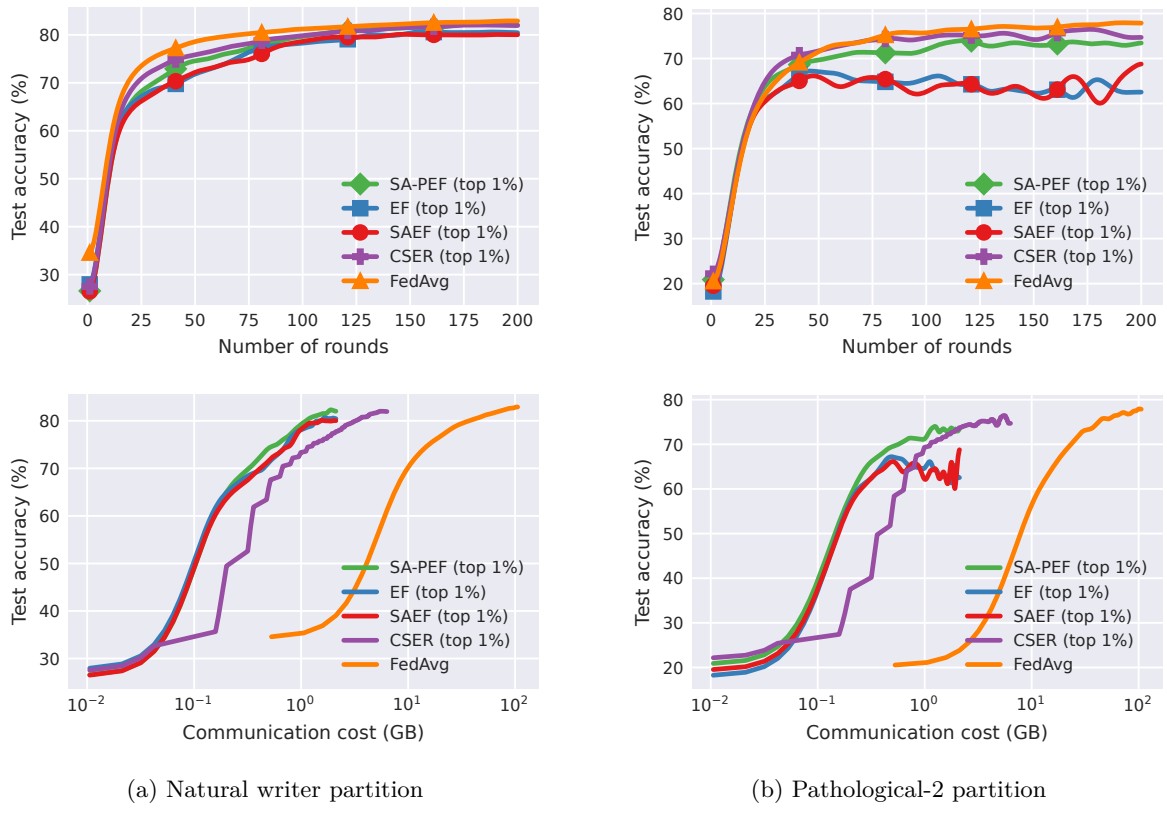

(a) Natural writer partition

(b) Pathological-2 partition

Figure 16: Test accuracy vs. number of rounds (row 1) and communicated GB (row 2) on FEMNIST.

The left column of Figure 16 shows the training curves under the natural writer partition. In this setting, SA-PEF is the strongest compressed method and remains competitive with dense FedAvg. The advantage over EF and SAEF is modest but consistent, suggesting that under natural writer heterogeneity, the benefit of partial step-ahead correction is real but limited.

To stress-test the methods under more severe label skew, we also construct a pathological-2 partition on FEMNIST, where each client is restricted to two classes. This split is substantially harsher than the natural writer partition and more directly exposes residual accumulation under aggressive compression. The right column of Figure 16 shows the corresponding training curves. In this regime, the difference between SA-PEF and the EF/SAEF endpoints becomes much larger: EF and SAEF both degrade markedly relative to dense FedAvg, whereas SA-PEF recovers a substantial portion of this gap. This behavior is consistent with the theory: as heterogeneity and residual accumulation become more pronounced, the trade-off controlled by $\alpha_r$ becomes more consequential, and the advantage of partial step-ahead correction becomes more apparent.

Overall, the FEMNIST results reinforce two observations. First, on a writer-partitioned federated benchmark, SA-PEF remains competitive and is the strongest compressed method among those we compare. Second, under a more extreme non-IID partition on the same dataset, the advantage of SA-PEF over EF and SAEF becomes substantially larger. This supports the main claim of the paper: the benefit of partial step-ahead correction is most evident in regimes where biased compression and client heterogeneity amplify residual-induced gradient–update mismatch.

