# OpenReview forum: "SA-PEF: Step-Ahead Partial Error Feedback for Efficient Federated Learning"
_TMLR — Accepted by TMLR_

### Review · Reviewer_uzSg · 2026-03-04

**Summary Of Contributions:**

## Summary

This paper proposes Step-Ahead Partial Error Feedback (SA-PEF), a generalized variant of error feedback for biased gradient compression in federated learning (FL). Building on standard error feedback (EF) and step-ahead EF (SAEF), the method introduces a tunable step-ahead coefficient $ \alpha_r \in [0,1] $, which previews only a fraction of the accumulated compression residual while leaving the remainder in the classical EF recursion. The theoretical analysis characterizes a step-ahead-controlled residual contraction factor and predicts an optimal range of $ \alpha_r $. Empirical results across multiple models, datasets, and compressors demonstrate faster convergence in communication rounds compared to standard EF.

---

## Strengths

- The paper clearly presents the proposed SA-PEF mechanism and positions it as a principled interpolation between EF and SAEF.
- The theoretical analysis is reasonably comprehensive, covering nonconvex objectives, δ-contractive compressors, data heterogeneity, and partial client participation.
- The derivation of a residual recursion and the identification of a step-ahead-controlled contraction factor provide insight into the observed early-phase acceleration.
- The experimental evaluation spans multiple architectures, datasets, and compressors, and consistently demonstrates improved communication efficiency over standard EF.

---

## Weaknesses

- Several important concepts and technical terms are introduced without sufficient preliminaries or formal definitions, which makes parts of the paper difficult to follow.
- The overall contribution appears somewhat incremental relative to SAEF, as SA-PEF mainly introduces a tunable interpolation parameter rather than a fundamentally new mechanism.

**Audience:**

Yes

**Audience Explanation:**

The topic is on federated learning, which is important and practical.

**Broader Impact Concerns:**

I don't think there are any broader impact concerns.

**Claims And Evidence:**

Yes

**Claims Explanation:**

The theoretical results and the simulation results look correct to me.

**Requested Changes:**

1. **Need for clearer preliminaries on EF and SAEF.**
   The paper would benefit from a dedicated preliminaries section that formally introduces standard error feedback (EF) and step-ahead EF (SAEF), including their update rules and residual recursions. Currently, only high-level descriptions are provided in the introduction and related work. Since SA-PEF is presented as an interpolation between EF and SAEF, a precise algorithmic description of both baselines is essential for readers to fully understand the novelty and technical differences.

2. **Unclear definitions of key quantities.**
   Some important terms are mentioned without explicit definitions. For example, the “pre-round residual contraction factor” (Pages 2 and 5) should be formally defined in math when first introduced.

3. **Unclear motivation at the end of Page 4.**
   The argument in the last paragraph of Page 4 is difficult to follow. In particular:
   (i) The definition of the "local displacement term" is not clearly stated.
   (ii) The notation $ g_k(\cdot;\cdot) $ is used without a precise definition at that point.
   (iii) It is not sufficiently explained why the boundedness (or lack thereof) of this term is critical to the subsequent analysis or motivation for SA-PEF.

---

> ### Author Response · Authors · 2026-04-20
>
> We thank the reviewer for the careful reading and constructive feedback. We appreciate both the positive assessment of the paper and the concrete suggestions for improving clarity and sharpening the distinction from SAEF. We address each point below. For ease of review, all changes in the revised manuscript are highlighted in blue.
>
> **1. Need for clearer preliminaries on EF and SAEF.**
> We agree. Since SA-PEF is framed as an interpolation between standard error feedback (EF) and step-ahead error feedback (SAEF), the manuscript needed a clearer side-by-side presentation of these baselines before introducing our method. In the revision, we expanded the _Relation to EF and SAEF_ discussion in Section 3 and added a side-by-side comparison table in the same section. This table explicitly contrasts the local start point, the residual composition, and the per-round residual-contraction factor for EF, the full step-ahead endpoint ($\alpha_r=1$), and SA-PEF. We believe this makes both the technical relationship and the novelty of SA-PEF much easier to parse.
>
> **2. Unclear definitions of key quantities.**
> In the revision, we now define the recurring quantities at first use instead of relying on later context. In particular, the expanded Section 3 discussion now introduces the averaged residual energy $\bar E_r := \tfrac{1}{K}\sum_k |e_r^{(k)}|^2$, the local-work scale $s_r := \eta_r L T$, the residual recursion $\bar E_{r+1} \le \rho_r \bar E_r + B_r$, and the explicit SA-PEF contraction factor $\rho_r$ before these quantities are used downstream in the analysis. This also makes it easier to connect the algorithmic intuition in Section 3 to the formal residual recursion and convergence discussion in Section 4.
>
> **3. Unclear motivation at the end of Page 4.**
> Thank you for pointing this out. We agree that this paragraph was too compressed in the original submission. In the revision, we rewrote the motivation paragraph in Section 3 (_Motivation: residual-induced mismatch under biased EF_) to explicitly define the effective de-errored point $\tilde w_r := w_r - d_r$, introduce the local displacement term, and explain why controlling this term matters. We now state directly that this mismatch contributes to the second moment of the accumulated local update, which then feeds into the residual recursion and ultimately the final stationarity bound. This motivates previewing only a fraction of the residual: doing so reduces gradient--update mismatch without fully sacrificing the stabilizing carry structure of classical EF.
>
> **4. Several important concepts are introduced without sufficient preliminaries or formal definitions.**
> We agree with this overall assessment. Beyond the specific changes above, we revised the exposition more broadly to improve the notation flow, define recurring technical objects earlier, and make the EF/SAEF/SA-PEF relationship explicit in the algorithm section rather than only through high-level prose. We also strengthened the roadmap from the Section 3 intuition to the Section 4 analysis so that the reader sees earlier how the mismatch discussion, the residual recursion, and the final stationarity guarantee fit together.
>
> **5. The contribution appears somewhat incremental relative to SAEF.**
> We appreciate this perspective. At the mechanism level, SA-PEF is indeed an interpolation between EF and full step-ahead EF through the parameter $\alpha_r$. However, our contribution is not merely introducing that interpolation. Rather, it is showing that this interpolation is both meaningful and analyzable in the federated local-SGD regime with biased compression and partial participation, which is outside the scope of the original SAEF analysis.
>
> Concretely, prior SAEF analysis is developed in a classical distributed optimization setting with a single synchronous step ($T=1$), a shared iterate at which gradients are evaluated, bounded-gradient-type assumptions, and full participation. In contrast, our setting requires controlling: (i) multi-step local drift under non-IID data, (ii) the second moment of the accumulated local update rather than a single-point gradient, (iii) biased $\delta$-contractive compression applied to that accumulated update, and (iv) partial client participation. These differences are reflected directly in the revised theory: the manuscript now highlights the FL-specific residual recursion, the step-ahead-controlled contraction factor $\rho_r$, its optimizer $\alpha_r^\star$, and the partial-participation extension in Section 4 and the appendix. Therefore, we agree that the mechanism is lightweight, but we believe the theoretical extension and the federated characterization are non-trivial and constitute the main contribution.

---

### Review · Reviewer_eKnq · 2026-03-23

**Summary Of Contributions:**

This paper studies a gradient–update mismatch issue in error feedback (EF) under biased compression in federated learning. To address this, it proposes SA-PEF, which introduces a coefficient $\alpha_r$ to partially apply the residual in a step-ahead manner, effectively interpolating between standard EF and full step-ahead EF.

  The approach is simple and easy to integrate into existing pipelines, and the paper provides both theoretical guarantees and empirical results showing improved communication efficiency. The motivation is clear, and the method is lightweight and broadly applicable.

  That said, the idea feels somewhat incremental, as it can largely be interpreted as a continuum between two existing variants. In addition, the performance appears sensitive to the choice of $\alpha_r$, and the paper does not fully explore behavior under more extreme settings such as high heterogeneity or aggressive compression.

**Additional Comments:**

No additional comments beyond the points raised above.

**Audience:**

Yes

**Audience Explanation:**

The paper tackles communication efficiency and optimization stability in federated learning, both of which are central topics for the TMLR community. The proposed method is simple, general, and easy to adopt, making it relevant not only to researchers but also to practitioners working on distributed or compressed training.

While the contribution is not highly novel, the perspective on residual dynamics and early-stage convergence could still be of interest, especially for those studying communication-efficient optimization.

**Broader Impact Concerns:**

The work focuses on improving communication efficiency in federated learning, which is generally aligned with privacy-preserving and scalable machine learning.

There are no immediate ethical concerns specific to the proposed method. However, like most federated learning approaches, potential issues may arise from data heterogeneity across clients, which can introduce bias or fairness concerns. A brief discussion acknowledging these risks would strengthen the broader impact section.

**Claims And Evidence:**

Yes

**Claims Explanation:**

Overall, the claims are reasonably supported by the combination of theory and experiments. The convergence analysis is solid and covers practical federated learning settings, and the empirical results consistently show improvements over EF in communication efficiency.

However, the experimental section leaves a few gaps. In particular, there is limited analysis of how sensitive the method is to the choice of $\alpha_r$, and the evaluation does not fully stress-test the method under more challenging regimes (e.g., stronger heterogeneity or heavier compression). Because of this, the evidence feels somewhat incomplete.

**Requested Changes:**

### Critical revisions

  - Provide a more detailed analysis of the sensitivity to the step-ahead coefficient $\alpha_r$. In particular, practical guidance for tuning this parameter, or an adaptive strategy, would make the method more usable.

  - Strengthen the empirical evaluation under more challenging settings, such as higher data heterogeneity, more aggressive compression, or larger numbers of local update steps. This would help better establish robustness.

  ### Non-critical suggestions

  - Add ablation studies to better isolate the effect of partial residual preview compared to standard EF and full SAEF.

  - Clarify the intuition behind the residual contraction factor and explain its practical implications more directly.

  - Expand the discussion on when SA-PEF should be preferred over EF or SAEF in real-world scenarios.

---

> ### Author Response · Authors · 2026-04-20
>
> We thank the reviewer for the thoughtful and constructive feedback. We appreciate the positive assessment of the theory, experiments, and practical simplicity of the method. We address each concern below. For ease of review, all changes in the revised manuscript are highlighted in blue.
>
> **1. The idea feels somewhat incremental.**
> We appreciate this perspective. We agree that, at a high level, SA-PEF can be viewed as interpolating between EF and SAEF through a single parameter $\alpha$. However, we do not claim that the mechanism is sophisticated. Rather, we show that this interpolation becomes analytically meaningful in the federated local-SGD regime with biased compression, client drift, heterogeneity, and partial participation. This regime differs substantially from the original SAEF setting: clients perform multiple local SGD steps ($T>1$), the compressed object is the accumulated local update rather than a gradient at a common point, and the residual dynamics must be controlled under biased/contractive compression and subsampling. We revised the paper to make this distinction clearer in Section 3 and throughout Section 4.
>
> **2. The method appears sensitive to the choice of $\alpha$.**
> We agree that this is an important practical point. The revised manuscript now emphasizes the dedicated sensitivity analysis in Section 5.3 (Figure 3), where we sweep $\alpha$ from $0$ to $1$ and compare EF ($\alpha=0$), SAEF ($\alpha=1$), and intermediate SA-PEF values. The main takeaway is that performance is strong over a fairly broad high-$\alpha$ regime rather than being concentrated at a single finely tuned value. We also made the practical implication more explicit in the text: SA-PEF can be used with a robust default in the $0.8$--$0.9$ range in our experiments. In addition, the revised discussion now notes that the theory suggests the optimizer $\alpha_r^\star = 1/(1+12s_r^2)$ for the contraction factor, which motivates future adaptive schedules based on an online estimate of the local-work scale $s_r$.
>
> **3. The empirical evaluation should be strengthened under more challenging regimes.**
> We appreciate this suggestion and agree that stronger stress tests would improve the empirical section. In the revision, we made the challenging settings more visible and also added new experiments in a substantially harder federated benchmark. Specifically, the appendix now includes an _Evaluation on FEMNIST_ subsection with both natural writer partitioning and a harsher pathological-2 split. These results complement the existing aggressive-compression, heterogeneity, and partial-participation experiments, and better isolate the regimes in which SA-PEF is most helpful.
>
> **4. Add ablation studies to isolate the effect of partial residual preview compared to EF and SAEF.**
> We thank the reviewer for this suggestion. In the revised manuscript, we now make this ablation logic more explicit. In Section 5.3, we compare the two endpoints directly, EF ($\alpha=0$) and SAEF ($\alpha=1$), while sweeping the intermediate SA-PEF values. This isolates the effect of partial residual preview cleanly: moving away from EF improves early progress, while stopping short of full step-ahead avoids the weaker late-stage behavior seen at $\alpha=1$.
>
> **5. Clarify the intuition behind the residual contraction factor.**
> In the revision, we have strengthened this point in two places. First, in Section 3, we now introduce the residual recursion and the role of $\rho_r$ earlier, together with the new side-by-side EF/SAEF/SA-PEF comparison table. Second, in Section 4.3 and the special-case contraction discussion, we now make the intuition more direct: a smaller $\rho_r$ means faster decay of the residual energy, which in turn reduces mismatch accumulation under biased compression and helps explain the early communication-efficiency gains observed empirically.
>
> **6. Expand the discussion on when SA-PEF should be preferred over EF or SAEF.**
> We appreciate this point. In the revised manuscript, we now state this more plainly in the discussion. In short, SA-PEF is most useful when biased compression is aggressive enough that classical EF suffers from slow residual decay, but full step-ahead ($\alpha=1$) is too aggressive under local drift and heterogeneity. In that regime, partial preview provides a practical middle ground: it keeps the early acceleration benefit of step-ahead correction while retaining part of the EF carry that supports late-stage stability.
>
> **7. Briefly acknowledge fairness risks from heterogeneity in the broader impact section.**
> We agree and have added this discussion to the _Broader Impact_ section. In the revision, we now explicitly state that improved optimization and communication efficiency do not by themselves resolve fairness concerns under heterogeneous client data, and we now encourage reporting beyond average metrics, such as worst-client accuracy, subgroup disparities, and participation-conditioned performance.

---

> > ### Author Response · Authors · 2026-04-20
> >
> > We appreciate the reviewer's suggestions. We believe these clarifications and the strengthened stress-test results have materially improved the paper.

---

### Review · Reviewer_cFKF · 2026-04-10

**Summary Of Contributions:**

This work proposed a new algorithm for communication efficient federated learning. Naive gradient compressors often suffer from biased estimate. the paper suggested using a step ahead error feedback to calibrate the gradient before performing the compression. Based on the proposed method the paper proves the convergence of the solver and shows the residual contraction analysis. Empirical results also suggest strong performance on the utility-efficiency trade-off on CIFAR 10 and 100.

**Audience:**

Yes

**Audience Explanation:**

New communication efficient FL paper with theoretical guarantees will be interested to TMLR audience.

**Claims And Evidence:**

Yes

**Claims Explanation:**

The paper is solid in general. There is extensive theoretical analysis on the algorithm and the insights from the theory seem to be reflected from empirical results.

**Requested Changes:**

In general I think the paper is well written. I like the theoretical analysis of $\alpha^*$ and the uniform residual contraction analysis and these are very important for understanding the method. the major thing I would like to see is that the experiment only consider cifar 10 and 100 which are not naturally partitioned. I'm curious to see results on naturally partitioned non iid datasets like FEMNIST and Sent140. Otherwise I think the paper looks good.

---

> ### Author Response · Authors · 2026-04-20
>
> We thank the reviewer for the positive assessment of the paper, particularly for recognizing the value of the residual-contraction analysis and the uniform contraction results. We address the main evaluation concern below. In the revised manuscript, all changes are highlighted in blue for ease of review.
>
> **Evaluation on a naturally partitioned federated benchmark.**
> We agree that evaluating the method on a naturally partitioned federated benchmark strengthens the empirical study. To address this point, we added a new appendix subsection, _Evaluation on FEMNIST_, together with learning curves for both the natural-writer FEMNIST split and a pathological-2 split.
>
> The natural FEMNIST partition contains roughly 3,500 writers. Because the compressed baselines considered in our study maintain per-client residual states, simulating all writers simultaneously with a 6.6M-parameter CNN would incur substantial memory and computational overhead. Following common benchmark practice [1], we therefore subsample a fixed pool of 200 writers per run and sample 20 clients per round. This preserves the natural writer-partitioned federated structure while keeping the experiments computationally tractable.
>
> We use a common setup across methods: a CNN (Conv32-$5\times5 \rightarrow$ Conv64-$5\times5 \rightarrow$ FC2048 $\rightarrow$ FC62; 6.60M parameters), 200 communication rounds, SGD with momentum $0.9$, weight decay $10^{-4}$, batch size $32$, 5 local epochs, and Top-$k$ compression with $k=1%$ plus error feedback for the compressed methods (FedAvg remains dense). Results are reported as mean $\pm$ standard deviation over three seeds.
>
> On naturally partitioned FEMNIST, SA-PEF remains competitive with dense FedAvg and is the strongest compressed method among those we compare. In particular,
>
> $$
> \text{SA-PEF: } 82.90 \pm 0.07 \text{ final accuracy}, \qquad
> 83.84 \pm 0.22 \text{ best accuracy},
> $$
>
> compared with
>
> $$
> \text{FedAvg: } 82.93 \pm 0.79 \text{ final accuracy}, \qquad
> 83.04 \pm 0.37 \text{ best accuracy}.
> $$
>
> By comparison, EF and SAEF achieve $81.63 \pm 0.57$ and $82.26 \pm 0.08$ final accuracy, respectively, while SA-PEF also attains the best average accuracy over the last 20 rounds ($82.59 \pm 0.05$).
>
> **Harder non-IID stress test on FEMNIST.**
> To further examine robustness, we also added results on a pathological-2 FEMNIST split, in which each client is restricted to two classes under the same communication and optimization settings. This split is substantially harsher than the natural writer partition and more clearly exposes the effects of residual accumulation under aggressive compression.
>
> In this regime, the gap between SA-PEF and EF/SAEF becomes much larger:
>
> $$
> \text{FedAvg: } 80.32 \pm 0.65,\quad
> \text{EF: } 67.88 \pm 4.43,\quad
> \text{SAEF: } 70.24 \pm 3.86,\quad
> \text{SA-PEF: } 77.03 \pm 3.77
> $$
>
> in final test accuracy. Relative to FedAvg, EF trails by roughly 12 percentage points and SAEF by roughly 10 percentage points, whereas SA-PEF reduces this gap to about 3.3 percentage points. We believe these results support the main claim of the paper: the benefit of partial step-ahead correction becomes most apparent in more challenging heterogeneous regimes, where residual accumulation and gradient-update mismatch are more severe.
>
> Overall, the new FEMNIST results support two conclusions: (i) on a naturally partitioned federated benchmark, SA-PEF remains competitive with dense FedAvg and is the strongest compressed method among those we compare; and (ii) under a substantially more extreme non-IID split on the same dataset, the advantage of SA-PEF over EF and SAEF becomes markedly larger.
>
> [1] Chen, D., Gao, D., Kuang, W., Li, Y., & Ding, B. (2022). pFL-bench: A comprehensive benchmark for personalized federated learning. _Advances in Neural Information Processing Systems_, 35, 9344--9360.

---

> > ### Comment · Reviewer_cFKF · 2026-04-29
> >
> > Thank you for your response. The results look good to me. Strongly suggest to put that in the main body of the paper

---

> > > ### Author Response · Authors · 2026-05-05
> > >
> > > Thank you for the positive feedback. We will move the key FEMNIST plot into the main experimental section, while keeping the full setup and supplementary results in the appendix.

---

### Decision · Action_Editor_9pmF · 2026-05-05

**Recommendation:** Accept as is

**Audience:**

Yes

**Audience Explanation:**

The study explores communication efficiency and optimization stability in federated learning, topics that are highly relevant to the TMLR community.

**Claims And Evidence:**

Yes

**Claims Explanation:**

The paper provides both rigorous theoretical and empiricla (even though somewhat limited) evidence to support the study of the proposed error-feedback technique.